# *Toxoplasma gondii* F-actin forms an extensive filamentous network required for material exchange and parasite maturation

Javier Periz[1†], Jamie Whitelaw[1†], Clare Harding[1†], Simon Gras[1], Mario Igor Del Rosario Minina[1], Fernanda Latorre-Barragan[1], Leandro Lemgruber[1], Madita Alice Reimer[1], Robert Insall[2], Aoife Heaslip[3*‡], Markus Meissner[1*]

[1]Wellcome Centre for Molecular Parasitology, Institute of Infection, Immunity & Inflammation, University of Glasgow, Glasgow, United Kingdom; [2]Cancer Research United Kingdom Beatson Institute, Bearsden, United Kingdom; [3]Department of Molecular Physiology and Biophysics, University of Vermont, Burlington, United States

*For correspondence: aoife.
heaslip@uconn.edu (AH); markus.
meissner@glasgow.ac.uk (MM)

[†]These authors contributed equally to this work

Present address: [‡]Department of Molecular and Cell Biology, University of Connecticut, Storrs, United States

Competing interests: The authors declare that no competing interests exist.

**Abstract** Apicomplexan actin is important during the parasite's life cycle. Its polymerization kinetics are unusual, permitting only short, unstable F-actin filaments. It has not been possible to study actin in vivo and so its physiological roles have remained obscure, leading to models distinct from conventional actin behaviour. Here a modified version of the commercially available actin-chromobody was tested as a novel tool for visualising F-actin dynamics in *Toxoplasma gondii*. Cb labels filamentous actin structures within the parasite cytosol and labels an extensive F-actin network that connects parasites within the parasitophorous vacuole and allows vesicles to be exchanged between parasites. In the absence of actin, parasites lack a residual body and inter-parasite connections and grow in an asynchronous and disorganized manner. Collectively, these data identify new roles for actin in the intracellular phase of the parasites lytic cycle and provide a robust new tool for imaging parasitic F-actin dynamics.

## Introduction

*Toxoplasma gondii* is a wide-spread obligate intracellular parasite that is thought to infect over two billion people worldwide. *T. gondii* infection of healthy individuals causes no major complications, infection can cause severe disease in immunocompromised individuals and foetuses infected *in utero*. The pathological manifestation is caused in a large part by repeated rounds of the parasite's lytic cycle, beginning with active invasion of the host cell by the parasite, replication within a specialized vacuole termed the parasitophorous vacuole (PV), followed by egress and lysis of the host cell. Replication occurs via a unique process of endodyogeny, where two daughter parasites are constructed within the mother, before elongating and budding, leading to the breakdown of the maternal parasite (*Francia and Striepen, 2014*). Remnants of the mother cell remain at the posterior end of the daughter parasites in a structure known as the residual body (RB), which has a role in organizing the parasites into their characteristic rosette pattern within the PV (*Muñiz-Hernández et al., 2011*; *Hu et al., 2002*). Membrane connections persist between the RB and the parasites until host cell lysis and parasite egress, presumably to allow inter-parasite communication (*Muñiz-Hernández et al., 2011*). To date little is known about the molecular mechanisms underlying these

**eLife digest** *Toxoplasma gondii* is a parasite that commonly infects most warm-blooded animals and is thought to affect over two billion people worldwide. In most cases, the infection does not cause any symptoms, although it can lead to serious complications in pregnant women or people with a weakened immune system.

*T. gondii* has a complex life cycle that involves different stages. During infection, the parasite invades the host cells and replicates inside a specialized cell structure called a 'parasitophorous vacuole' until the host cell bursts. The parasite then spreads and infects more host cells. The replication is synchronised, meaning all parasites in a host cell replicate at the same time. It was unclear how the parasites coordinated this process, but some researchers suggested that the parasites remained connected to each other to communicate by exchanging material and information. A good candidate to form such connections is the protein actin, which in many organisms forms filaments that guide the transport of cargo molecules in the cell. However, previous research indicated that actin in *T. gondii* is incapable of forming these stable filaments.

Periz et al. developed a new tool of fluorescence markers that specifically bind to actin in *T. gondii* and found extensive actin networks that connected parasites with each other and also to the membrane of the parasitophorous vacuole. Actin was needed to transport molecules between the parasites within a vacuole and was also found to enter the cells of the parasite. When the protein was depleted in the parasite, the network collapsed; the parasites started to replicate at different times and could no longer leave the host cell.

A next step will be to further investigate the role of actin in *T. gondii* and other parasites using the tools developed by Periz et al. A better understanding of replication of *T. gondii* could provide clues to new treatments for parasitic diseases that cause substantial economic losses worldwide.

processes. Previous studies focused on the role of microtubules during parasite replication, since treatment with microtubule inhibitors leads to structural collapse of the parasite, while actin-modulating drugs are thought to cause only slight defects in replication (*Shaw et al., 2000*). In contrast, research on parasite actin focused on its crucial role during host cell invasion and egress (*Dobrowolski and Sibley, 1996*). However, recent studies have also implicated actin and myosins in intracellular processes including apicoplast division (*Egarter et al., 2014*; *Andenmatten et al., 2013*; *Jacot et al., 2013*), secretory organelle (dense granule) transport (*Heaslip et al., 2016*) and parasite replication (*Haase et al., 2015*).

Actin is a highly conserved protein, which forms dynamic filaments in eukaryotic cells. Through an association with actin-binding proteins, these filaments are themselves organized into higher order structures that play important roles in a wide variety of cellular functions including muscle contraction, vesicle transport and cytokinesis. *T. gondii* actin is encoded by a single gene, *act1* and has only ~80% sequence identity with mammalian actin isoforms but shares 93% similarity with *Plasmodium* ACT1 (*Dobrowolski et al., 1997*). Apicomplexan ACT1 is clearly essential, and compared to its counterparts in higher eukaryotes is believed to be intrinsically unstable, resulting in the formation of only short filaments (*Skillman et al., 2011*). Biochemical assays indicate that 97% of the parasites actin is present in the globular form (*Dobrowolski et al., 1997*; *Skillman et al., 2011*; *Wetzel et al., 2003*). It has been proposed that apicomplexan actin is unique amongst actins as it polymerizes in a highly unusual, isodesmic manner (*Skillman et al., 2013*). According to the isodesmic polymerisation model, monomer addition is governed by a single equilibrium constant, meaning that no (unfavourable) activation step is required to initiate the formation of the first dimer leading to polymerisation. In this instance, nucleation and elongation are equally favourable. This contrasts to cooperative polymerisation, where the activation step is the formation of the first dimer/trimer, which has a higher equilibrium constant than polymer elongation (*Smulders et al., 2010*). Therefore polymer formation can only occur above a critical concentration (Cc) of monomers (*Pantaloni et al., 1985*). It is this activation step that is regulated by actin nucleators, such as the Arp2/3 complex or formins (*Carlier et al., 2015*). Puzzlingly, formins and other actin nucleating proteins have been shown to have essential roles in *Toxoplasma* and *Plasmodium*, begging the question of their function if they

are not required to initiate actin polymerisation or accelerate filament elongation (*Baum et al., 2008*; *Jacot et al., 2013*). A recent study suggested that the polymerization process of apicomplexan actin needs to be reinvestigated, as heterologously expressed apicomplexan actin, the basis for many of the previous studies, is incorrectly folded (*Olshina et al., 2016*). Furthermore, it was demonstrated that conditional deletion of *act1* in *T. gondii* results in complete abrogation of known actin functions, long before G-actin levels are fully depleted, suggesting that in vivo the formation of F-actin depends on a critical monomer concentration (*Whitelaw et al., 2017*). This raises concerns about previous studies of actin polymerization kinetics based on mis-folded actin. Furthermore, imaging studies on different life cycle stages of the apicomplexan parasite, *Plasmodium falciparum,* demonstrated the formation of an extensive F-actin cytoskeleton in both gametocytes and ookinetes (*Hliscs et al., 2015*; *Siden-Kiamos et al., 2012*). Using 3D-SIM it was demonstrated in gametocytes that these filaments appear to be organized in the cytoplasm, below the Inner Membrane Complex (IMC; a specialised structure found in apicomplexans that consists of membranous vesicles and structural components located just beneath the plasma membrane) of the parasite rather than between the IMC and the plasma membrane (*Hliscs et al., 2015*), where it is thought to power parasite gliding motility (*Meissner et al., 2013*).

To further address the role of ACT1 in parasite growth, we characterized a recently generated conditional actin knockout parasite line (*act1 cKO*) during the intracellular portion of the parasites lytic cycle (*Andenmatten et al., 2013*). We demonstrate that *act1* cKO parasites lack a residual body and grow in an asynchronous and disorganized manner within the PV. Further investigation of actin functions required imaging the filamentous actin cytoskeleton. Previous attempts to visualise F-actin within the parasites have largely been unsuccessful, since conventional actin labelling techniques such as Life-Act, Phalloidin and SiR-Act do not allow detection of F-actin within the parasites. GFP-actin shows a high signal to noise ratio in parasites causing the inability to differentiate actin filaments from the monomeric form *Angrisano et al. (2012a)*. Thus we sought a new approach to image F-actin in intracellular parasites by expressing actin-chromobody, a single chain anti-actin camel antibody that has been successfully employed in diverse eukaryotes, including plants (*Rocchetti et al., 2014*) and animal cells (*Plessner et al., 2015*; *Panza et al., 2015*). Chromobodies (Cb) were found to have several advantages compared to other actin probes, such as lower toxicity, less influence on F-actin dynamics and a high signal to noise ratio (*Panza et al., 2015*). Using this probe, we were able to visualise filamentous actin in live parasites. While some filamentous structures were seen within the parasite cytosol as expected, we were surprised to observe extensive F-actin networks in the RB and F-actin-containing membranous tubules linking parasites within a vacuole. These tubules appear to be involved in the transport of material between parasites and recycling of the mother IMC at the end of the replication cycle. Collectively, these data identify new roles for actin in the intracellular phase of the parasites lytic cycle and provide a robust new tool for imaging parasitic F-actin dynamics.

## Results

### Depletion of actin results in the loss of the residual body

We previously characterised a conditional knockout for *Toxoplasma* actin (*act1 cKO*) and found that, in addition to its important role during gliding motility and host cell invasion, parasite actin is essential for maintenance of the apicoplast, dense granule motility and host cell egress (*Egarter et al., 2014*; *Heaslip et al., 2016*). Additionally, a recent study demonstrated a role for actin in parasite replication (*Haase et al., 2015*) and thus we used the *act1 cKO* to further examine the role of actin in parasite growth. Although initially intracellular *act1 cKO* parasites replicate at comparable rate to wild-type parasites (*Egarter et al., 2014*), they grew asynchronously and appeared disorganised, without the characteristic rosette organization of parasites in the parasitophorous vacuole (*Figure 1A and B*). While parasites replicate normally up to the 4 cell stage, later divisions are not tightly synchronised (*Figure 1A*), meaning that within the same PV parasites can be identified that are at different stages of endodyogeny (*Figure 1B*, arrow). In addition, the intravacuolar network, as visualised using Gra2 antibodies (*Mercier et al., 1993*), appeared disorganised and malformed (*Figure 1A*). Using scanning electron microscopy (SEM) of intracellular parasites, we observed that in wild-type (RH) *T. gondii* a membranous network connects parasites at their posterior pole whereas

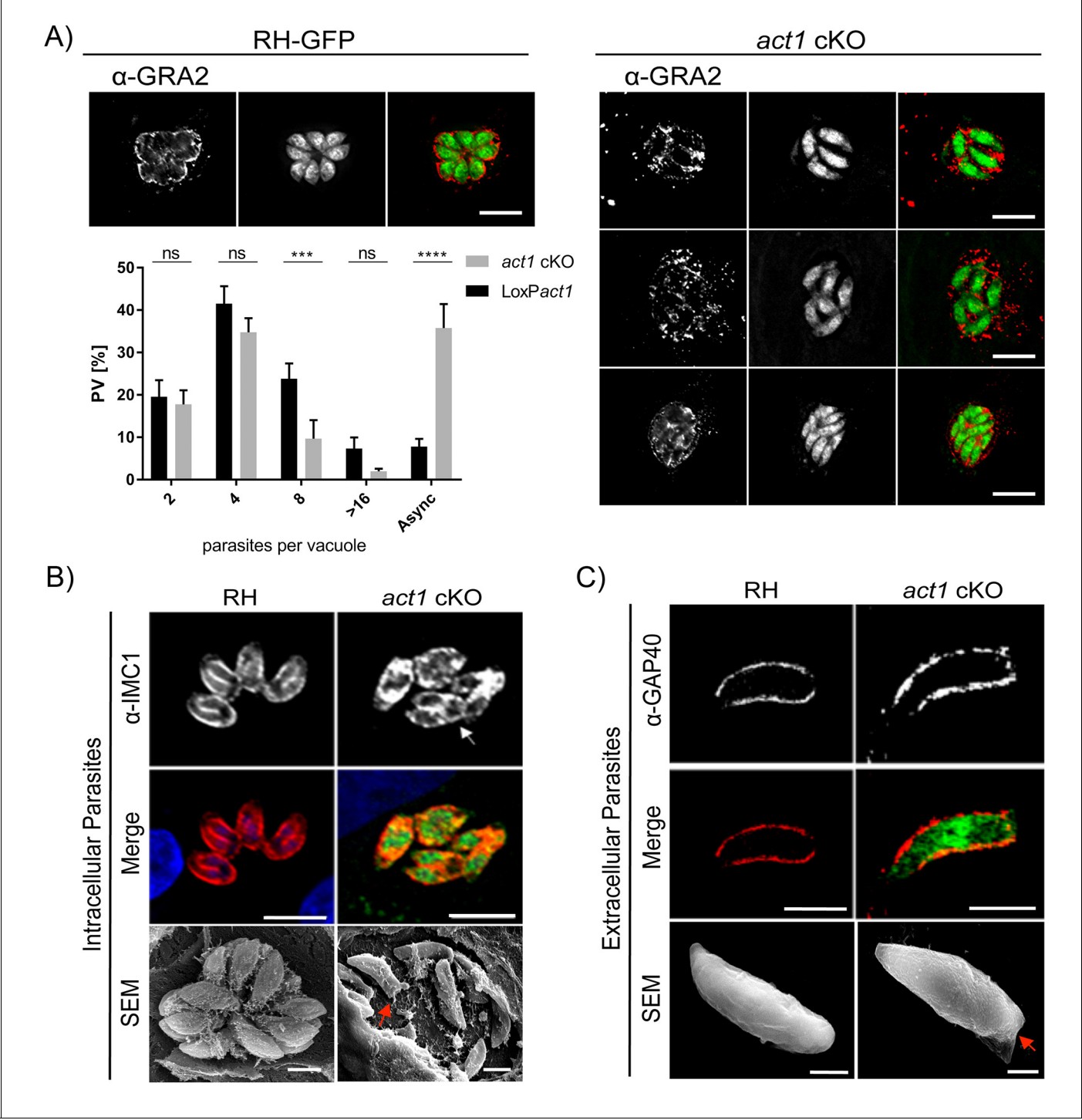

**Figure 1.** Analysis of intracellular development of *act1* cKO. (A) Conditional null mutants for *act1* were induced as previously described in order to excise *act1* (*Egarter et al., 2014*). 96 hr post induction, parasites were fixed and stained with anti-Gra2 to visualise the intravacuolar network. Scale bars: 10 µm. Replication assay of loxP*act1* and *act1* cKO. Number of nuclei per parasitophorous vacuole were counted 24 hr after inoculation on HFF cells. Mean values of three experiments in triplicate are shown. Asynchronous division is indicated by PVs with unusual amounts of parasites ($\neq 2^n$). Two-way ANOVA followed by Sidak's test were used to compare means between groups. ****p<0.0001, ***p<0.001, non-significance (ns) p>0.05. For source data see supplemental information (*Figure 1A—source data 1*). (B) *Top image*: The developing IMC of parasites is shown using IFA. Note that in case of *act1* cKO individual parasites are in different stages of replication (i.e. white arrow indicates a parasite at the end of replication). Scale bars: 10 µm. *Bottom image*: Scanning EM analysis in intracellular parasites. Scale bars: 2 µm. Note the disordered appearance of parasites within the PV in absence of ACT1 and flattened bottom of the *act1* KO parasites (Red Arrowhead). (C) Analysis of the IMC of the *act1* cKO using IFA and scanning EM

*Figure 1 continued on next page*

*Figure 1 continued*

analysis in extracellular parasites. *act1* cKO parasites have a flattened bottom, torpedo shape (Red arrowhead). The posterior pole of the parasite appears to be misformed, indicating a role of ACT1 during the final stages of replication (see also *Figure 9*). Scale bars: fluorescence images: 5 μm, SEM: 2 μm.

The following source data is available for figure 1:

**Source data 1.**

no obvious connections between individual parasites were observed in the *act1* cKO (*Figure 1B*). In addition, imaging of extracellular parasites by SEM and immunofluorescence showed aberrant morphology at the posterior pole in the *act1* cKO parasites, which appears generally flattened (*Figure 1C*, arrow). Together, these data suggest a role of actin in the formation of the RB, the organization of parasites within the PV and the intravacuolar network. Furthermore, asynchronous replication of parasites within the PV may indicate a loss of signalling between individual parasites.

## Actin is required for material transport between individual parasites within the parasitophorous vacuole

Based on electron microscopy evidence, the RB has been predicted to be involved in the transfer of material between parasites within a vacuole (*Muñiz-Hernández et al., 2011*) although this has not been previously demonstrated. To investigate this hypothesis and the role of actin in this process, we infected host cells with wt or *act1* cKO parasites expressing GFP and bleached individual parasites within the PV before measuring the time of fluorescence recovery (*Videos 1* and *2*; *Figure 2A*). While the fluorescence signal recovered rapidly in wt parasites (~30 s), no fluorescence recovery could be observed in parasites depleted of actin, indicating a defect in material transfer (*Figure 2A*). From this experiment, we concluded that individual parasites remain connected via the intravacuolar network and can exchange cytoplasmic material. Furthermore, these connections appear to be actin dependent.

To determine if vesicular as well as cytoplasmic material could be transferred between parasites, we performed time-lapse analysis of the integral membrane protein GAPM1a-YFP (*Figure 2B*) and identified vesicles moving in a directed manner, demonstrating that vesicular transport occurs outside of the parasite (*Figure 2B*, *Video 3*). Of note, digital tracking of vesicles suggested a directional movement along a tubular or filamentous structure.

In summary, these data indicate that individual parasites within a vacuole remain connected via a network, possibly through the RB that is required to transfer both cytoplasmic and membrane bound material between individual parasites. We hypothesize that parasite actin is required for the formation and/or maintenance of this interconnecting network.

## Expression of chromobodies in *T. gondii*: Identification of a filamentous actin network connecting parasites within the parasitophorous vacuole

We hypothesised that actin may be involved in residual body formation and inter-parasite communication. However, F-actin has not previously been visualised in live parasites, complicating a functional characterisation of F-actin and F-actin dynamics in apicomplexan parasites. In order to visualize F-actin in *T. gondii*, we modified a commercially available actin-chromobody (Cb) that has been established as a novel tool to study F-actin dynamics in living cells. This single chain camel antibody specifically recognizes F-actin and has been successfully employed to study actin dynamics in diverse eukaryotes. We generated two expression vectors for Cb, where Cb is fused to either Emerald GFP (Cb-EmGFP; [*Shaner et al., 2007*]) or Halo (Cb-Halo; [*Los et al., 2008*]). Upon transient expression of these proteins, we obtained identical staining of filamentous structures within the parasites. Cb was also localised somewhat diffusely in the parasite cytosol, probably corresponding to unbound Cb in the cytosol as the protein was expressed at a high level (*Figure 3A*). Strikingly, individual parasites within the PV remain connected by a filamentous actin network which stretches throughout the RB and parasitophorous vacuole and can be seen to enter the posterior pole of individual parasites (*Figure 3A*, arrow). With increasing size of the parasitophorous vacuole, an impressive intravacuolar network consisting of F-actin became apparent (*Figure 3B*) that appeared to not

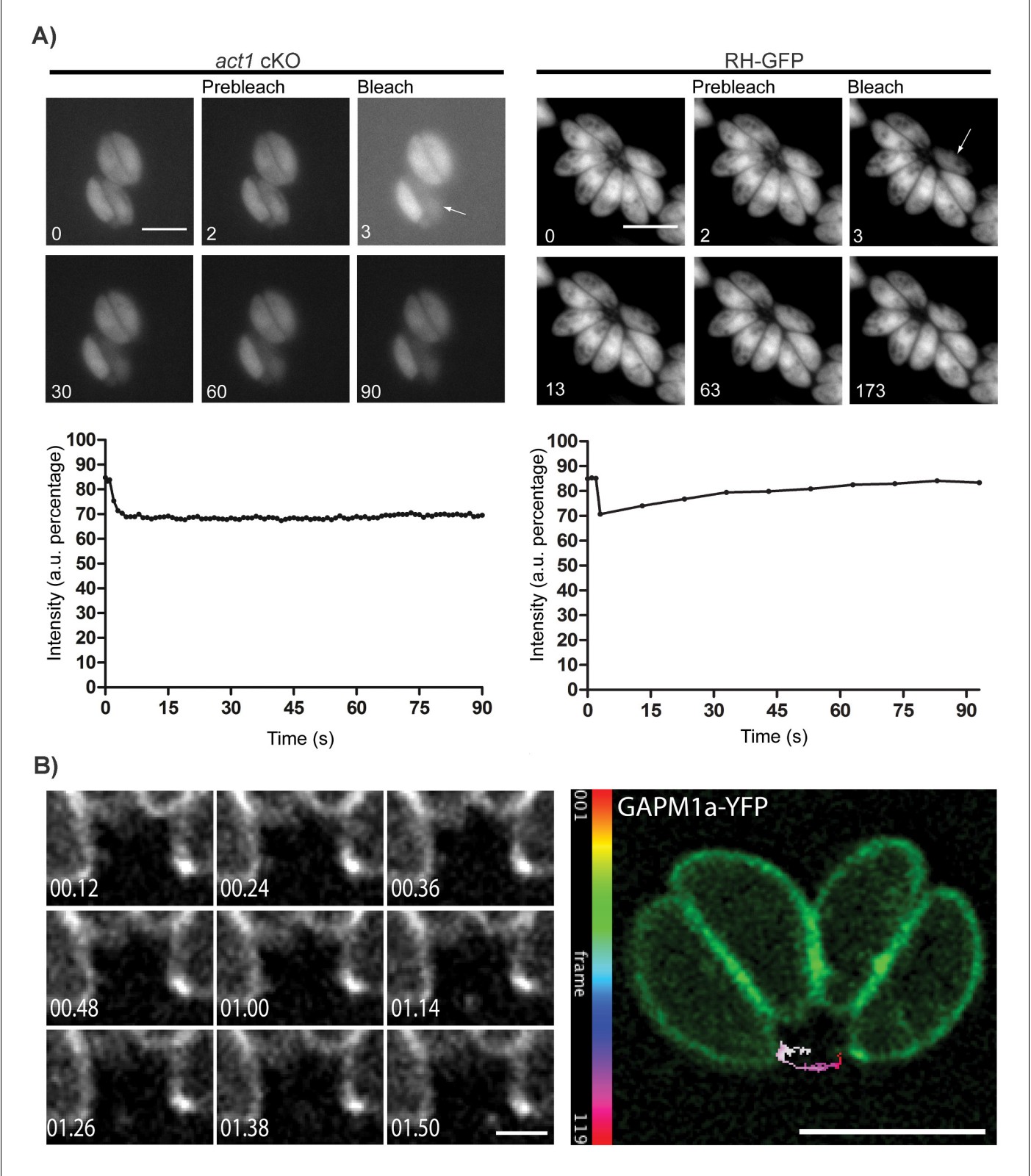

**Figure 2.** Material transport in between individual parasites within a PV. (**A**) FRAP treatment in a conditional *act1* cKO 72 hr after excision of *act1* (Left panel). FRAP treatment shows no recovery in fluorescence intensity in the bleached cell (arrow) over the duration of the experiment (90 s), indicating that the absence of actin abrogates transport of constitutively expressed YFP between neighbouring cells. FRAP treatment in a wt strain constitutively expressing GFP shows recovery in a bleached cell as soon as 20 s after bleaching (right panel, control experiment). Intensity in the FRAP area was

*Figure 2 continued on next page*

*Figure 2 continued*

expressed as intensity percentage of the same area in a cell unbleached. Scale bar; 5 μm. See also *Videos 1* and *2*. FRAP experiments shown are representative for several biological replicates (n > 3). (**B**) Extracellular vesicles are actively transported in wt parasites. Vesicles positive for GAPM1a-YFP were seen to be transported. Individual vesicles were tracked and their path color coded by frame according to indicated scale. Time is indicated in minutes. Scale bar: 5 μm, detail 1 μm.

only connect individual parasites, but also reached from the centre to the edge of the PV (*Figure 3B*, Figure 7).

During transient expressions of the fusion proteins, the majority of intracellular parasites appeared healthy with no apparent morphological changes, indicating that expression of Cb-proteins is well tolerated by the parasite. To analyse the effects of Cb expression on actin dynamics, we used dense granule motility as a surrogate marker, since it has been shown to critically depend on actin dynamics (*Heaslip et al., 2016*). Parasites stably expressing SAG1-ΔGPI-GFP (to label the dense granules) were transiently transfected with Cb-Halo and dense granule motility was analysed and correlated to mean fluorescent intensity of Cb-Halo within the parasite (*Video 4*; *Figure 3C*). Strong expression of Cb-Halo leads to an almost complete block of dense granule motility, however, weaker expression levels are well tolerated with no significant change in either dense granule run frequency (*Figure 3C*) or run length compared to controls (935 ± 39 nm vs 1013 ± 53 nm for control and Cb-Halo, respectively).

In summary, expression of Cb is well tolerated by the parasite indicating that it does not significantly adversely affect normal actin functions within the cell, in agreement to data obtained in other cellular systems (*Belin et al., 2014*; *Panza et al., 2015*). However, it cannot be ruled out that actin dynamics are locally affected within the cell due to expression of Cb.

## Generation of stable parasite lines expressing Cb-Halo or Cb-Emerald

As transient transfection resulted in a heterologous population of parasites expressing Cb at varying levels, which may be deleterious for the parasites, we generated stable parasite lines expressing either Cb-Halo or Cb-Emerald and confirmed that parasite growth, invasion, replication and egress rates are indistinguishable from wt parasites (*Figure 4A–E*). To ensure minimal influence of Cb

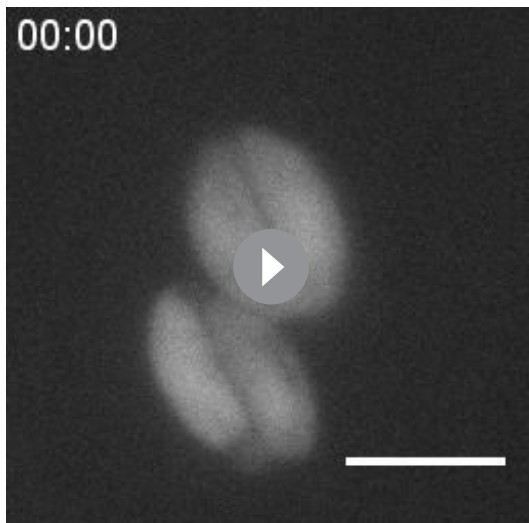

**Video 1.** FRAP on *act1* cKO. FRAP treatment in a conditional *act1* cKO 72 hr after excision of *act1* in loxPAct1. After FRAP treatment in a cell in the PV, no recovery in fluorescence intensity was observed. Scale bar 5 μm. Imaging speed 5 fps.

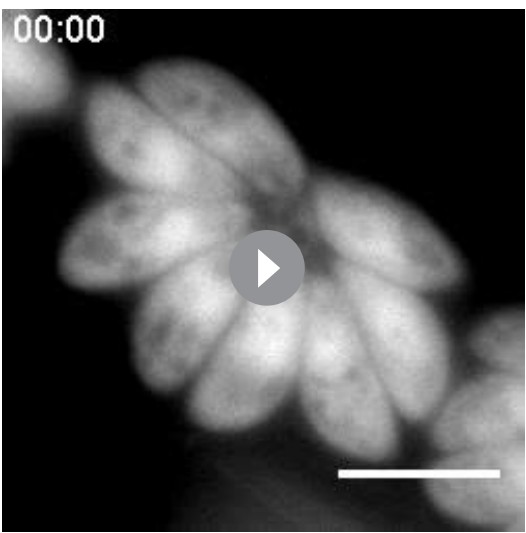

**Video 2.** FRAP on RH-YFP. FRAP treatment in a RH strain constitutively expressing GFP shows recovery in a bleached cell after a period of 20 s. Scale bar 5 μm. Imaging speed 5 fps.

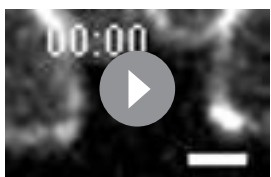

**Video 3.** Vesicular motility of GAPM1A-YFP in parasites endogenous expressing GAPM1a-YFP vesicles positive for GAPM1a-YFP were seen to be transported extracellularly. A vesicle was tracked and its path indicated. Time is indicated in minutes, scale bar 1 µm.

expression on actin dynamics in our stable line, we also analysed gliding motility, an important process depending on F-actin dynamics (*Dobrowolski and Sibley, 1996*; *Drewry and Sibley, 2015*; *Jacot et al., 2013*; *Opitz and Soldati, 2002*; *Skillman et al., 2011*; *Wetzel et al., 2003*). We found that the rate of overall motility is slightly increased (p<0.05) upon expression of Cb (*Figure 4F*), indicating an influence of Cb expression on actin dynamics. Intriguingly, average trajectory lengths are identical in wt and Cb expressing parasites (*Figure 4G*), while average gliding speed is reduced (*Figure 4H*). We also analysed average invasion speeds and confirmed that parasites penetrate the host cell within 30 s irrespective of Cb expression (*Figure 4D*). Furthermore, time lapse imaging of Cb-Emerald demonstrated a highly dynamic behaviour of F-actin within the cytosol (*Video 5*). Together these data demonstrate that, similar to the situation in other eukaryotes, the expression of this actin binding protein has no or modest effects on established actin-dependent processes in *T. gondii*. Moreover, as we identified little phenotypic consequence due to stable Cb expression in actin function, this reagent will be a useful tool for detecting parasite F-actin in live cells, and characterizing actin dynamics and organization.

## Cb specifically binds to parasite F-actin and does not alter the total amount of F-actin

To further validate Cb for its use in apicomplexan parasites, we next analysed binding characteristics of Cb to F-actin. Since apicomplexan actin cannot be functionally heterologously expressed (*Olshina et al., 2016*), we used skeletal chicken actin to estimate the binding characteristics of Cb to F-actin. Recombinant Cb with a C-terminal 6-His tag (rCb) was expressed and purified from bacteria (*Figure 5A,B*). rCb bound to chicken skeletal actin with $K_d$ of 5 ± 1 µM (*Figure 5B*). To confirm that Cb-Halo binds specifically to *Toxoplasma* F-actin when expressed in the parasite, we co-immunoprecipitated actin from extracellular parasites using an anti-Halo antibody (*Figure 5C*). As expected, only a small proportion of actin was precipitated, which corresponds well with published data suggesting that only ~2% of total parasite actin can be found in its filamentous form in extracellular parasites (*Dobrowolski and Sibley, 1997*). Importantly, mass-spectrometric analysis of the immunoprecipitation confirmed that Cb-Halo specifically precipitated parasite actin. Although additional, potential F-actin binding proteins were detected, no host cell actin, actin related or actin-like proteins could be identified (not shown). To assess if expression of Cb stabilises F-actin in parasites, we compared the amount of pelletable actin in parasites expressing Cb and wt parasites (*Figure 5D*). We confirmed that in both cases F-actin is barely detectable in extracellular parasites, indicating that it is primarily found in globular form. Treatment of parasites with Jas allowed detection of significant amount of F-actin in the pellet. Of note, no significant difference could be observed between control parasites and parasites expressing Cb. Together, these data confirm that expression of Cb in parasites allows detection of F-actin and that Cb expression has only minor effects on F-actin dynamics, as previously shown in other eukaryotes (*Panza et al., 2015*).

## Cb positive structures correspond to parasite F-actin

Next, we wished to determine if Cb binds specifically to F-actin within the parasites. Using actin-modulating drugs, the actin cytoskeleton was either depolymerized with Cytochalasin D (Cyt-D) or stabilized with Jasplakinolide (Jas). Treatment of intracellular parasites with Cyt-D led to the disintegration of the inter-parasite connections with only punctate spots remaining, which were also identified with anti-actin antibody (*Figure 6A*). In contrast, treatment with Jas led to the stabilisation of F-actin and intra-parasite connections were easily detectable with either Cb or an anti-actin antibody (*Figure 6A*). Using default settings of the Ridge Detector plugin (*Steger, 1998*) we determined the apparent number, maximum length and average filament size of PVs treated with Jas, Cyt-D or untreated PVs (control) (*Figure 6A* right panel). In Jas-treated PVs, and in contrast to Cyt-D treated

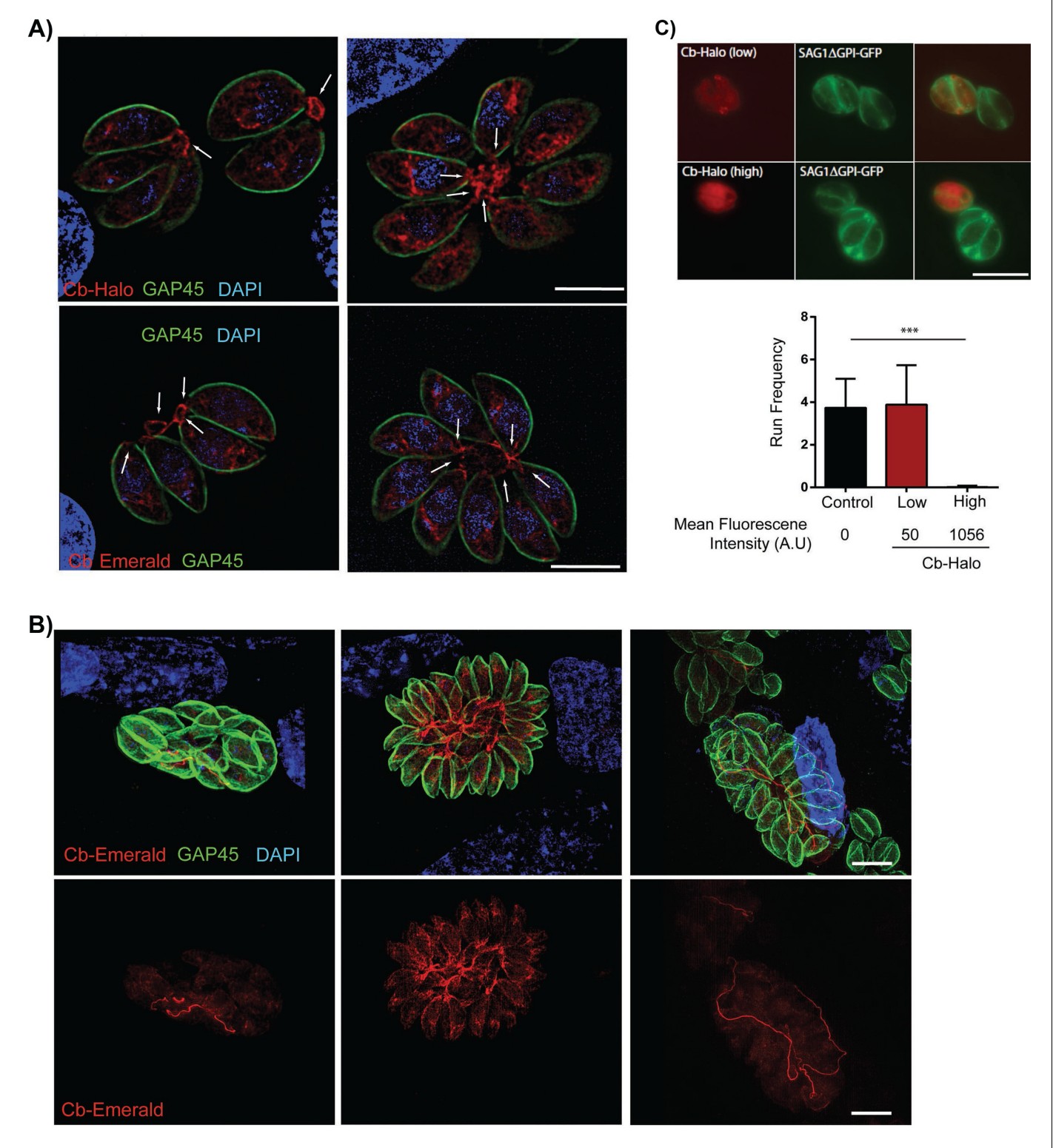

**Figure 3.** Filamentous actin can be visualized by expression of Cb-Halo and Cb-Emerald. (**A**) Sections of parasite vacuoles in two, four, and eight-cell stage stably transfected with Cb-Halo or Cb-Emerald (red). A filamentous network connecting individual cells (arrows) can be visualised using Cb-Halo and Cb-Emerald. Parasites were co-stained with IMC marker, GAP45 (green), DAPI staining (blue) Scale bar 5 µm. (**B**) Images of larger parasite vacuoles containing 16 or 32 parasites transfected with Cb-Emerald (red). Parasites were co-stained with IMC marker GAP45 (green). Note the formation of the extensive intravacuolar network with long filamentous tubes (see also *Figure 7*). Scale bar 10 µm (**C**) *Top.* Image of parasites expressing SAG1ΔGPI-
*Figure 3 continued on next page*

*Figure 3 continued*

GFP to label the dense granules and low and high levels of Cb-Halo. Scale bar 10 μm. *Bottom.* Directed granule run frequency in control (non-expressing) parasites and parasites expressing Cb-Halo at high and low levels. ***p<0.0001; students t-test. Total number of directed runs counted in control, low expression and high expression samples were 183, 150 and 1 respectively. Total number of vacuoles analysed from control, low expression and high expression were 19,17 and 14, respectively, from two independent transfections. Error bars are S.E.M.

PVs, there is an increase in average and maximum filament size, together with a decrease in total number of short filaments supporting the stabilisation and depolymerisation of Cb-Halo detected filaments in the presence of Jas and Cyt-D respectively. An intermediate situation occurs in untreated PVs with long filaments and high number of short filaments supporting a dynamic regulation in actin polymerisation in control samples. Overall, these results support that filaments detected with Cb-Halo behave as typical actin filaments in the presence of actin regulatory drugs. This supports the evidence that Cb-Halo is specifically labelling F-actin filaments and that Cb binding to actin does not prevent depolymerisation caused by Cyt-D.

To confirm that parasite F-actin corresponds to Cb-positive structures, we expressed Cb-Halo in *act1 cKO* cells. Using these parasites, we found that as early as 24 hr after excision of the *act1* gene, the filamentous network collapsed, leaving punctate actin spots visible only in the RBs (similar to vacuoles treated with Cyt-D). By 48 hr onwards, no F-actin structures could be observed and Cb-Halo was completely cytosolic, as expected due to cytosolic expression of this reagent (*Figure 6B*). The loss of filaments within the *act1* cKO over-time follows the down-regulation of ACT1 in *act1* cKO, indicating a polymerisation mechanism similar to eukaryotic actins (*Whitelaw et al., 2017*). To exclude a role of host cell actin in the formation of the inter-parasite connections, we treated infected host cells with the actin-disrupting drug latrunculin A, which specifically inhibits polymerization of host cell, but not parasite, actin (*Hegge et al., 2010*; *Whitelaw et al., 2017*). We found that latrunculin A treatment for 3 hr led to complete disruption of host cell F-actin (visualised using Phalloidin$_{488}$), while the parasite F-actin network remained unaffected. This confirms that Cb-positive filamentous structures were not derived from host cell actin (*Figure 6C*). Next we wished to compare F-actin dynamics in a conditional mutant for the actin depolymerisation factor (ADF; [*Mehta and Sibley, 2011*]). Previous studies demonstrated that the knockdown of ADF leads to stabilisation of actin filaments (*Mehta and Sibley, 2011*) and apicoplast loss (*Haase et al., 2015*). When we expressed Cb-Emerald in *adf* cKD to compare actin dynamics, we found that in absence of anhydrotetracycline (ATc) - when ADF is expressed - parasite F-actin shows a highly dynamic behaviour similar to wt parasites (*Figure 6D*, compare *Videos 6* and *7*). In stark contrast, incubation of parasites with ATc for 96 hr leads to depletion of ADF and consequently F-actin dynamics is significantly diminished. Under these conditions F-actin filaments accumulate at the posterior and to a lesser extent at the apical pole of the parasites. However, no dynamic behaviour can be detected and no filaments can be detected within the cytosol of the parasites (*Figure 6D* (red arrows), *Videos 6* and *7*).

## Inter-parasite actin tubules are dynamic during parasite replication and egress

As we had now established that endogenously expressed Cb bound specifically to parasite F-actin, we sought to investigate the dynamics of the inter-parasite F-actin network during the parasite's life cycle. Using time-lapse analysis, we found that in non-replicating parasites F-actin forms an extended, continuous filamentous network through the RB, connecting the parasites. During parasite replication, this network collapses and F-actin retreats to the RB (asterisks) before the network reform again,

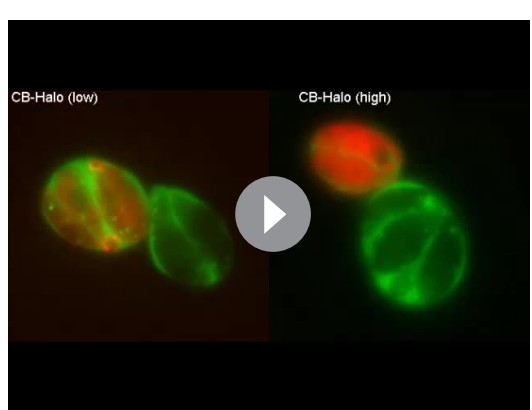

**Video 4.** Dense Granule dynamics in intracellular *T. gondii* parasites expressing SAG1△GPIGFP and low or high levels of CB-Halo. Imaging speed 10 fps, playback 6x real time.

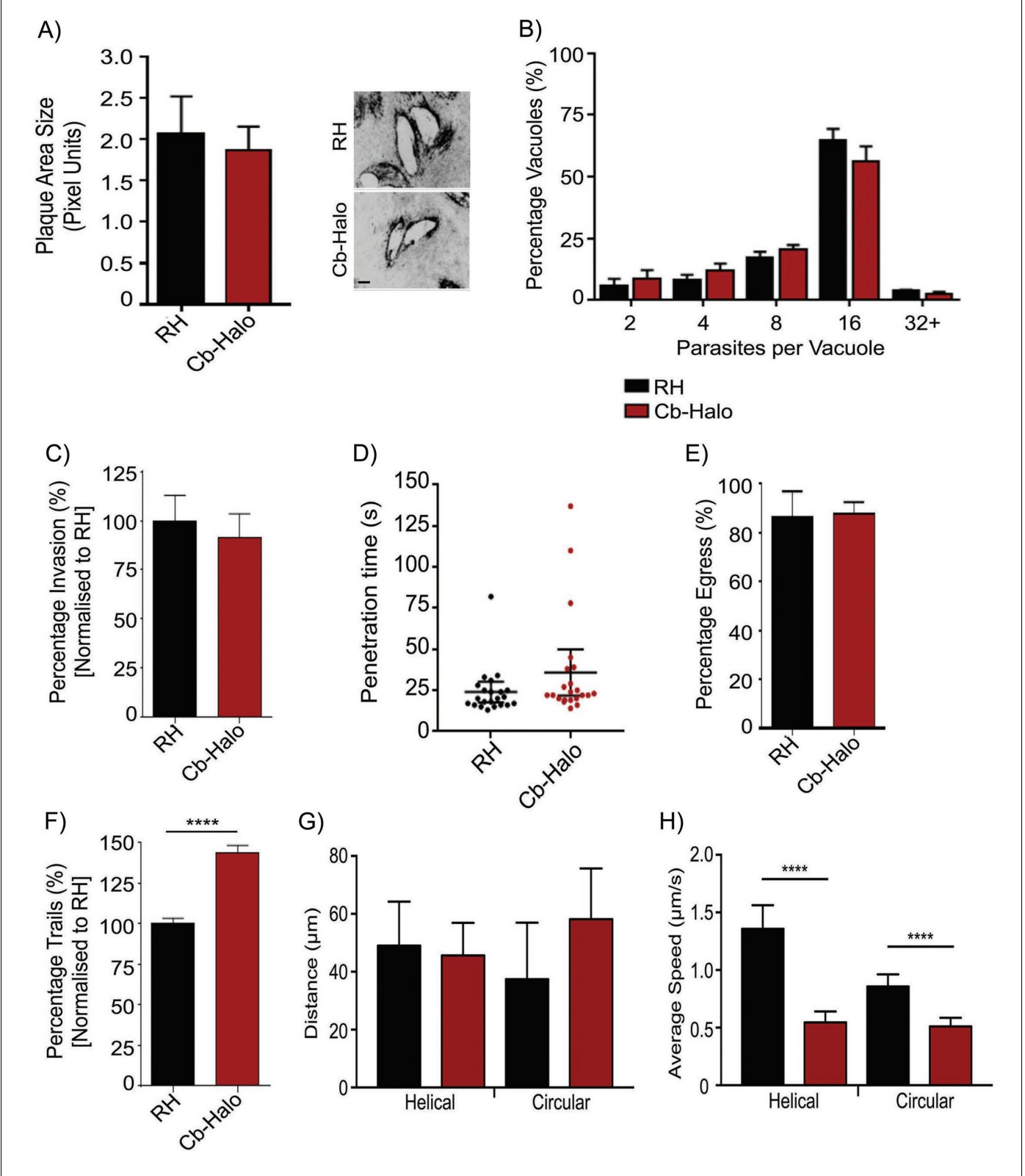

**Figure 4.** Phenotypic analysis of parasite expressing Cb-Halo. (**A**) *Left.* Growth assay of indicated parasites. After 5 days growth no significant difference in growth rates can be observed. *Right.* Representative image of plaque area size created. Scale bar, 100 µm. (**B**) Replication rates between Cb-Halo and RH parasites are comparable. Indicated parasites were inoculated on HFF cells and number of parasites/vacuole was determined. (**C**) Invasion rates are not significantly different between RH and Cb-Halo expressing parasites. Parasites were allowed to invade for 1 hr, before non-invaded parasites

*Figure 4 continued on next page*

*Figure 4 continued*

were removed. (**D**) Penetration time of parasites was determined using time-lapse analysis. Both RH and Cb-Halo expressing parasites are capable to invade the host cell within 30 s. In some cases slower parasites can be detected, although on average the difference is insignificant. n = 22 independent events. (**E**) No differences in egress could be detected between RH and Cb-Halo parasites. Egress was triggered using Calcium Ionopore A23187 and the number of egress events was determined after 10 min. (**F**) Trail deposition assay comparing gliding rates between RH and Cb-Halo. Cb-Halo parasites form slightly more trails (*p<0.05). (**G,H**) Comparison of gliding motility between RH and Cb-Halo. Whereas average run length (**G**) is identical, parasites expressing Cb-Halo demonstrate slower gliding speeds for helical and circular motility. Parasites were tracked with Fiji wrMTrck software. N = 20 individual events for each condition. All assays were conducted in triplicates. Datasets were compared using two-tailed Student's t-test. Error bars for A,B,C,E,F represent S.E.M from three independent, biological replicates. Error bars for D, G, H represent 95 % CI. *p<0.05, ****p<0.0001. For source data see supplementary information (*Figure 4—source data 1*).

The following source data is available for figure 4:

**Source data 1.**

---

extending throughout the PV (*Figure 7A*, *Video 8*). Note that these filaments appear relatively static in resting parasites and do not show much reorientation/movement within 8 hr. However, once parasites start to replicate within the PV, the connections disintegrate and F-actin appears to be restricted to the RB (*Figures 7A* and *8A*, *Videos 8* and 12).

As the parasites appeared to be connected by F-actin filaments within a vacuole, we wondered how the organization of this network changes as the parasite exits from the host, especially as *act1* cKO parasites are unable to egress (*Egarter et al., 2014*). When we triggered a calcium signalling cascade using Ca$^{2+}$-Ionophore (*Black et al., 2000*), the F-actin network collapsed rapidly, between 10 and 60 s after Ca$^{2+}$-Ionophore addition (*Figure 7B*, *Video 9*). Collapse of the network preceded the initiation of motility and egress from the host cell (*Figure 7B*, arrow head; *Video 9*). As the parasites begin to move away from the lysed host cell, F-actin can be detected at the rear of the parasites (*Figure 7B*, inset) and residual filamentous actin is still seen within the RB.

To investigate the presence of F-actin within this network, we performed correlative light-electron microscopy (CLEM). The network within a vacuole was imaged using 3D structural illumination super-resolution microscopy (3D-SIM) (*Figure 7C*). Thin sections of the same network were then imaged with transmission electron microscopy (squares). This demonstrated that extracellular F-actin filaments reside within membranous tubules of 50–60 nm in diameter. Within these tubules, several ~5 nm thick filaments (arrows) extending over 100 nm in length were observed (highlighted in magenta). Taken together these results confirm the presence of bundles of actin filaments bound within a membranous network, which connects individual parasites within the PV. This situation appears very similar to the formation of tunnelling nanotubes, long filopodia like structures, which consists of thin F-actin-based membranous structures with a small diameter (20–500 nm) that facilitate long range communications between cells (*Abounit and Zurzolo, 2012*).

Given that the inter-parasite tubules are reorganized during both replication and egress, we investigated the dynamic behaviour of F-actin within individual tubules using fluorescence recovery after photobleaching (FRAP). While F-actin dynamics within the parasite cytosol is very fast (*Video 10*) and recovery rates very rapid, within 20 s (*Figure 7D*), we found that

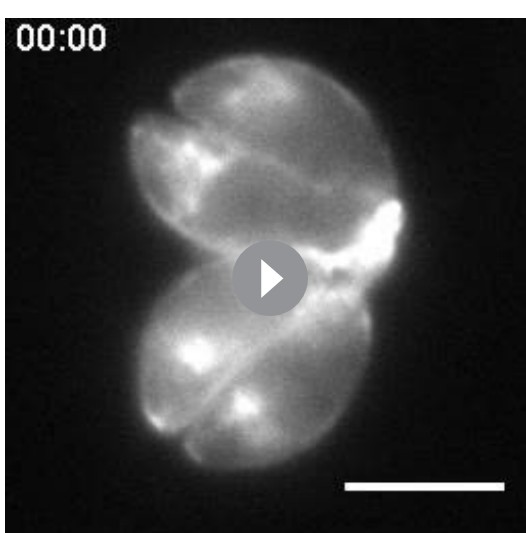

**Video 5.** Parasites expressing Cb-Emerald demonstrate highly dynamic F-actin dynamics. After FRAP treatment in a cell in the PV, recovery of fluorescence intensity is observed due to polymerization of new actin filaments formed from the cell periphery to the bleached cytoplasm region. Scale bar 5 μm. Imaging speed 5 fps.

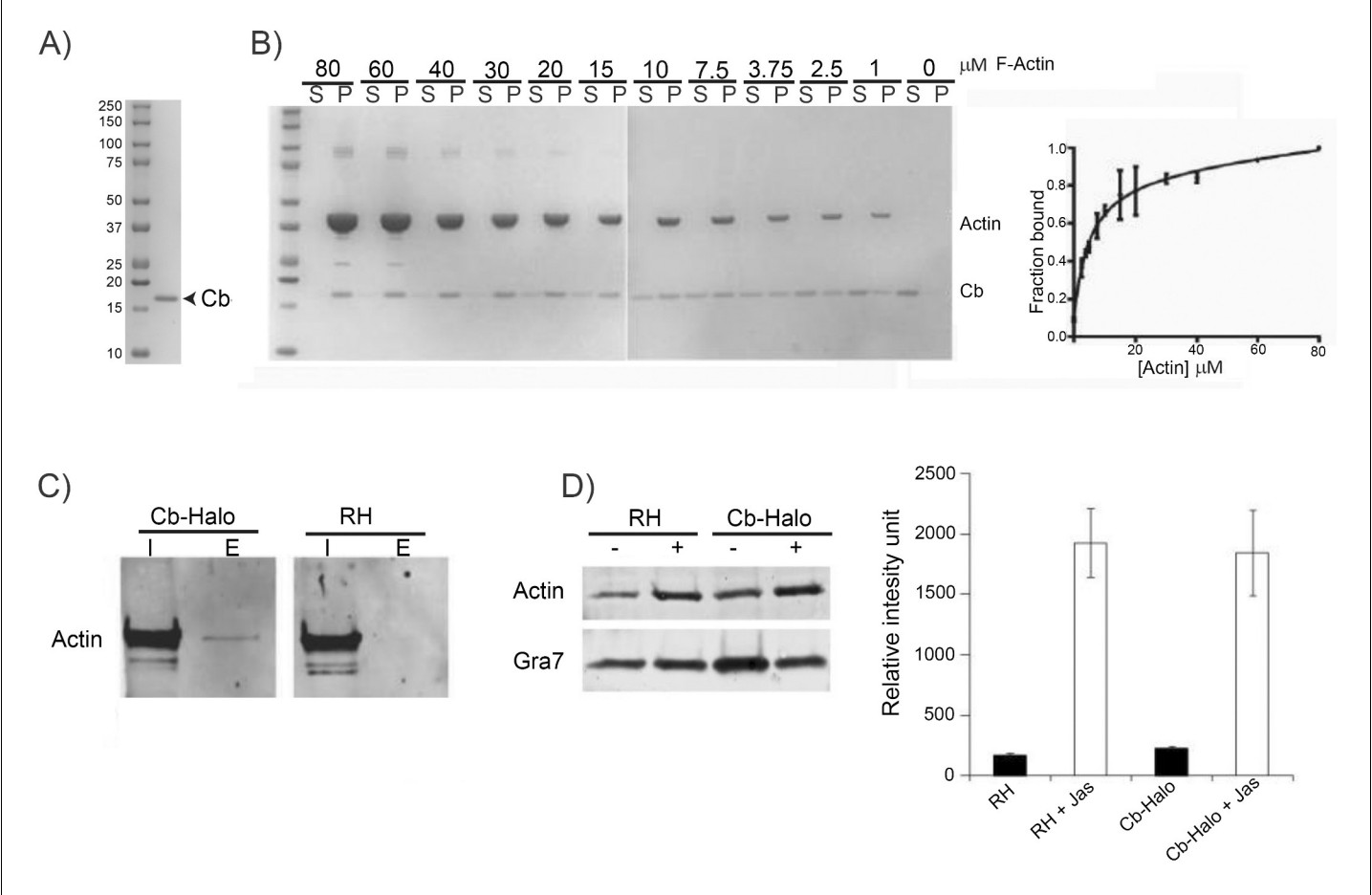

**Figure 5.** Chromobodies are specific for parasite F-actin and do not influence total amount of F-actin in the parasite. (**A**) Coomassie stained gel showing recombinant Cb purified from bacteria. (**B**) *Left.* rCb affinity assays. Coomassie stained gels showing in vitro binding of purified Cb (4 µM) to variable range of skeletal chicken F-Actin concentration (80 µM down to 0 µM). Supernatant (S) and pellet (P) were separated by ultracentrifugation. *Right.* Quantitative analysis. Ratio between purified Cb in the supernatant and pellet was determined. Solid line is a fit of the binding equation to the data ($Kd = 5 \pm 1.2$ mM). Results obtained from two independent experiments. (**C**) Interaction between Cb-Halo and actin in *T. gondii* Cb-Halo expressing strain. Western blot comparison of input lysate (I) and elution (E) obtained from co-immunoprecipitation using beads against the halo-tag with the Cb-Halo strain and RH. Actin pull-down was only detected in the Cb-Halo expressing strain. (**D**) Sedimentation assays. Actin sedimentation, with and without Jasplakinolide (1 µM) was evaluated for Cb-Halo strain and RH. GRA7 was used as loading control and signal intensity normalisation between conditions. Increased amount of F-actin was found in the pellet of parasites incubated in the presence of Jasplakinolide. However, no difference between RH and parasite expressing Cb-Halo could be detected in both control and Jas treated condition (n = 6).

F-actin within the tubular network is much more stable and fluorescence labelling of these structures took more than 60 s to fully recover (***Video 11***). This demonstrates the presence of highly dynamic, intracellular F-actin and a stable F-actin containing filamentous network (***Figure 7D***).

Finally, given that *act1* cKO parasites divided asynchronously we wanted to readdress the role of actin in parasite replication. We generated parasites co-expressing GAPM1a-YFP and Cb-Halo and performed live imaging (***Figure 8A***, ***Video 12***). During early stages of daughter cell formation, F-actin was found in the RB linking the two parasites at their posterior end (***Figure 8A***; 1.24–2.36 hr). At the earliest stages of daughter cell construction, F-actin accumulated at the elongating IMC (***Figure 8A***: 1.36–2.36 hr) and further concentrated towards the posterior end of the mother cells, where it colocalized with the mother IMC (***Figure 8A***, 3.00–3.24 hr). As the daughters begin to bud from the maternal cell, the IMC of the mother disintegrates and appears to be transported towards the daughter cells (***Figure 8A***, Arrows, 3.24–3.48 hr). At the end of replication, the now mature parasites remain connected through F-actin structures (***Figure 8A***, 4.00–5.12 hr).

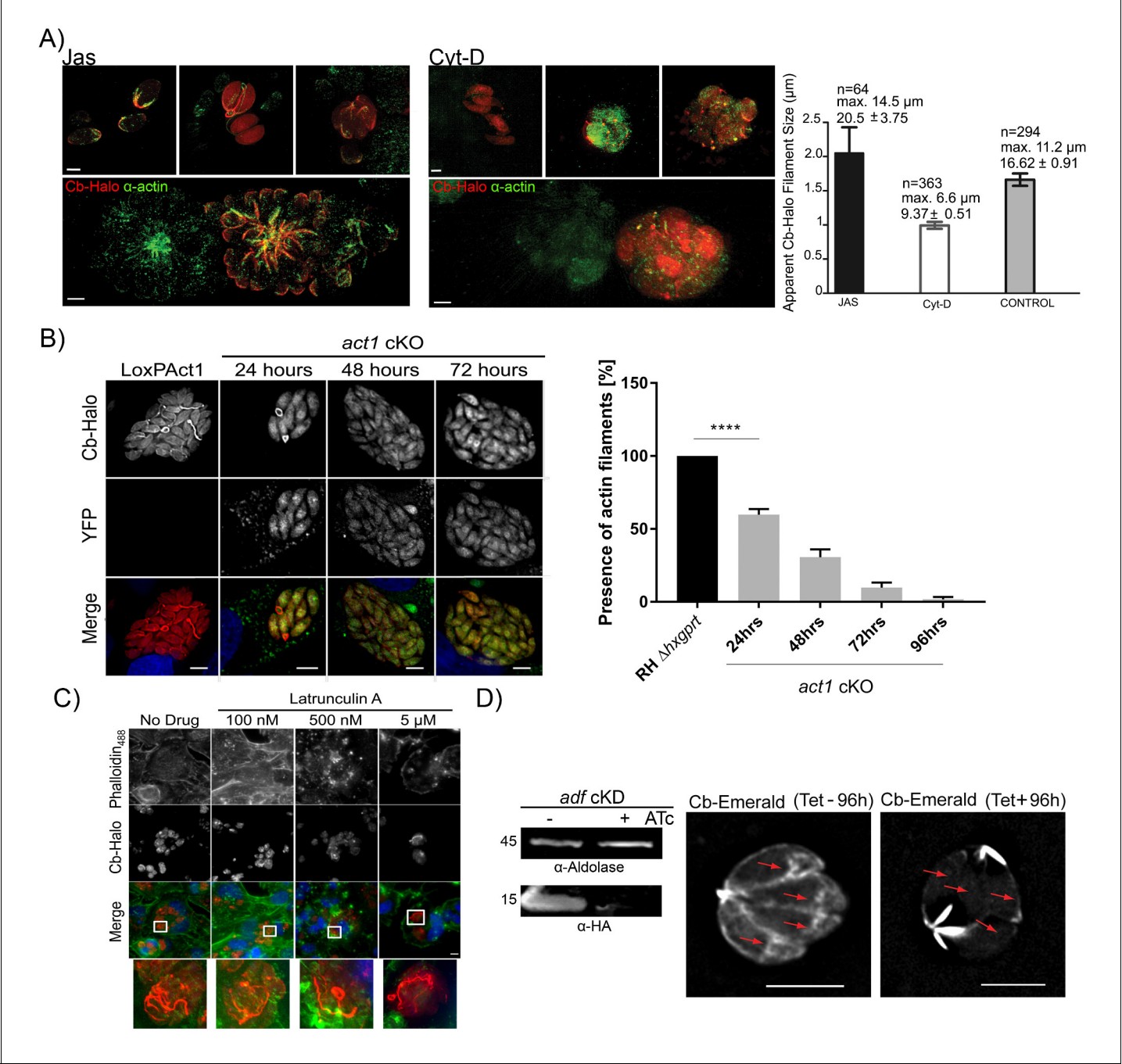

**Figure 6.** Chromobodies specifically stain parasite F-Actin. (**A**) 3D-SIM imaging of parasites transiently expressing Cb-Halo (red) and stained using actin antibody (green) (*Angrisano et al., 2012b*). Treatment with 100 nM Jas for 1 hr results in formation of an elaborate network, whereas treatment with 2 μM Cyt-D, the filamentous network collapses. Note that in the bottom images a field of view was selected were a transiently transfected PV expressing Cb-Halo is next to a non-transfected PV. Both show identical staining. Scale bars 2 μm. Right: Quantification of the apparent filament size in five representative PVs growth for 24 hr, as measured using default settings in the Ridge Detection Plug In (ImageJ, see Material and Methods), n is the number of filaments, max size is the longest filament found in each condition expressed in μm. Number below indicates average pixels +/- SEM (10 pixels correspond to 1 μm). (**B**) Expression of Cb-Halo in a conditional act1 cKO. A filamentous network is observed prior to excision of *act1*. As soon as 24 hr after excision of act1 the network diminishes. Right: The percentage of vacuoles with >8 parasites containing actin filaments in RH *Δhxpgrt* and *act1* cKO were quantified at 24, 48, 72 and 96 hr after induction. Mean values of three experiments in triplicate are shown. One way ANOVA followed by Dunnett's multiple comparisons test was used to compare means between groups. ***p<0.0001. (**C**) Host cell actin is not involved in formation of the filamentous network. Parasites were allowed to replicate for 24 hr, before being treated for 3 hr with indicated concentration of Latrunculin A. Host cell F-actin was visualised with Phalloidin488 (green). Scale bars; 10 μm. (**D**) Analysis of actin dynamics in a conditional mutant for *adf*. Left: Immunoblot
*Figure 6 continued on next page*

*Figure 6 continued*

using indicated antibodies. ADF-HA is depleted upon treatment of parasites with 1 µM ATc. Parasites were grown for 96 hr in HFF cells in the presence and absence of inducer, before being artificially released. Equal amounts of parasites were loaded. Right: Representative still image of *Video 6* and *7*. Parasites were grown for 96 hr in the presence or absence of ATc. Note that upon depletion of ADF no actin filaments can be detected in the cytosol of the parasites (arrows) and F-actin accumulates at the posterior and (to a lesser extent) apical pole of the parasite. Scale bar: 5 µm. For source data (A,B) see supplementary information (*Figure 6—source data 1*).

The following source data is available for figure 6:

**Source data 1.**

Given that GAPM1a-YFP positive vesicles appear to be transported between individual parasites within a PV or between parasites and the residual body (*Figures 2B* and *8A*) we wished to further define the role the filamentous network plays in this process. We imaged Cb-Halo and GAPM1a-YFP co-expressing parasites and tracked the motion of vesicles within the residual body. GAPM1a-YFP vesicles moved outside the parasite along filamentous tubules (*Figure 8B*), demonstrating vesicular transport within the residual body, as also shown for wt parasites (*Figure 2B*). Importantly, transport of vesicles was dependent on F-actin, as incubation of parasites with Cyt-D significantly abrogated vesicular transport (*Figure 8C*). In summary, these data demonstrate that individual parasites remain connected via an F-actin containing network that is required to transfer material in an active, F-actin dependent process (*Video 13*).

## Discussion

Studies on *Toxoplasma* actin performed in the 90's suggested that apicomplexan actin is highly divergent and incapable of forming stable filaments (*Dobrowolski and Sibley, 1997*; *Dobrowolski et al., 1997*; *Dobrowolski and Sibley, 1996*). Since then, polymerization kinetics of heterologously expressed apicomplexan actin showed that it polymerized in an unusual isodesmic process, in contrast to conventional eukaryotic actin (*Skillman et al., 2013*). These findings have been recently questioned, as heterologously expressed apicomplexan actin was shown to be misfolded (*Olshina et al., 2016*). In parallel, a number of recent genetic studies have demonstrated important functions of parasite actin during

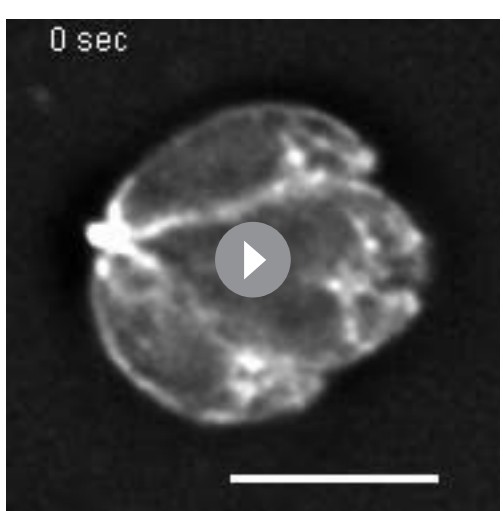

**Video 6.** Analysis of actin dynamics in *adf* cKO expressing Cb-Emerald in absence of ATc. Note the highly dynamic F-actin within the cytosol of the parasites. Scale bar 5 µm. Imaging speed 5 fps.

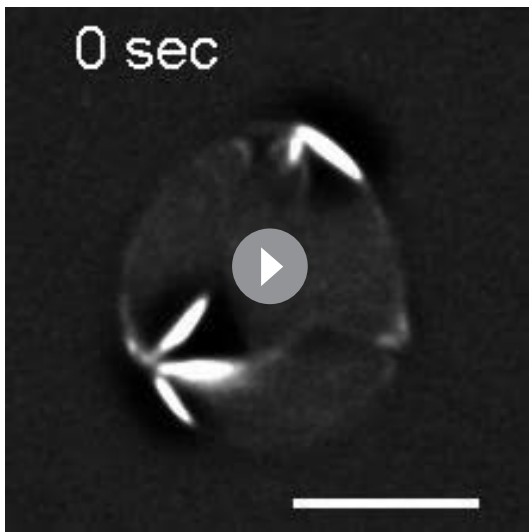

**Video 7.** Analysis of actin dynamics in *adf* cKO expressing Cb-Emerald in the presence of ATc (when ADF is depleted). Actin dynamics is almost completely abolished and F-actin can be found concentrated at the posterior end and much less at the apical tip of the parasite. Scale bar 5 µm. Imaging speed 5 fps.

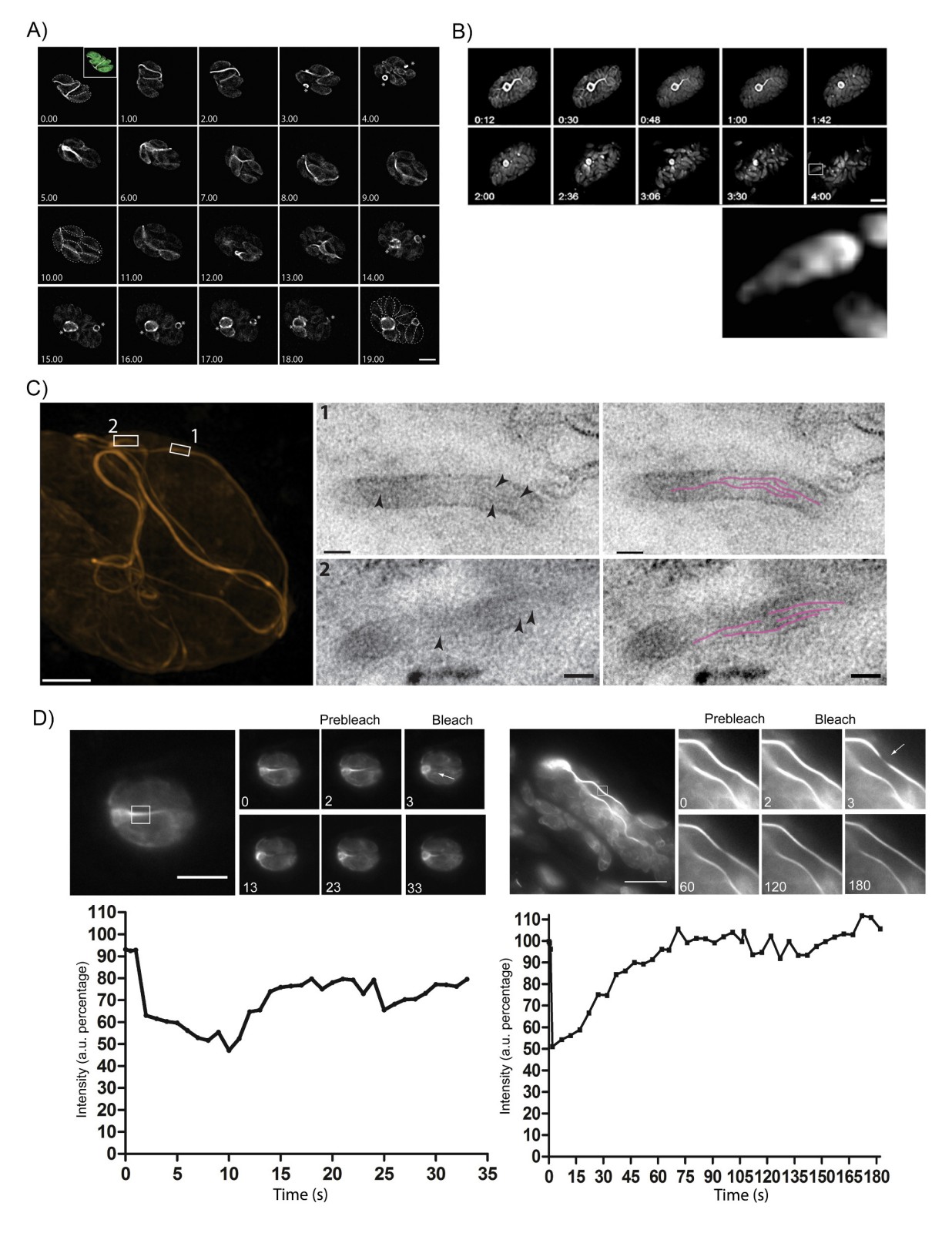

**Figure 7.** The F-actin network is stable in resting parasites, but highly dynamic during replication and egress. (**A**) Analysis of Cb-Halo during two rounds of replication. Images were taken every 30 min for 20 hr (*Video 8*). The network appears dynamic across the intracellular lifecycle, collapsing into rings during daughter cell emergence (asterisks). Time indicated in hours. Scale bar 5 μm. (**B**) Collapse of the F-actin network can be triggered by calcium signaling. Parasites were induced for egress with Calcium Ionophore A23187 and imaged at 1 frame per second (*Video 9*). The network collapses
*Figure 7 continued on next page*

*Figure 7 continued*

before parasites begin to egress. While filaments quickly collapse, the residual body remains intact during egress and is left behind. Box in lower left image shows freshly egressed parasites (enlarged below), where F-actin appears to accumulate at the posterior pole of the parasite. Time indicated in minutes:seconds. Scale bar, 10 μm. (C) Correlative light electron microscopy (CLEM). A vacuolar network was imaged with 3D-SIM super-resolution microscopy and the same areas were imaged with TEM. Filaments of 5 nm in thickness were present within the network tubules, extending over 100 nm in length. Scale bars: 200 nm (3D-SIM); 50 nm (TEM). (D) FRAP treatment in cells stably expressing Cb-Emerald. The F-actin inside the cells (left panel) and the nanotubular network connecting the parasites (right panel) show different fluorescence recovery times (20 and 60 s respectively), suggesting the presence of different actin dynamics inside the parasite and the filamentous network of the PVs respectively. Intensity in the FRAP area was expressed as intensity percentage of the same unbleached area (filament or nanotubular network). Time is expressed in seconds. Scale bar 5 μm.

intracellular development, including maintenance of the apicoplast (*Andenmatten et al., 2013*; *Egarter et al., 2014*), daughter cell replication (*Haase et al., 2015*) and motility of secretory organelles (*Heaslip et al., 2016*). These findings cannot be easily reconciled with the current view of parasite actin being incapable of efficient polymerization, and would instead predict the presence of F-actin filaments within the parasite cytosol. Furthermore, there are 11 putative myosin motors within the parasite (*Foth et al., 2006*), driving diverse cellular processes from cell division to motility, which require actin filaments to function.

A major impediment to resolving this controversy has been the lack of appropriate reagents for specifically labelling F-actin. Parasite actin does not bind phalloidin, the gold standard reagent in other eukaryotic systems (*Skillman et al., 2011*) and, attempts to use genetically encoded actin sensors, such a LifeAct (*Riedl et al., 2008*) or Utrophin-CH (*Burkel et al., 2007*) have failed thus far (our and others unpublished data). Using a novel tool based on camelid nanobodies, Chromobody, we have visualized F-actin in *T. gondii* for the first time in live parasites and demonstrated it has important and previously unforeseen roles during the intracellular development of the parasite. While individual actin filament kinetics could not be easily defined, due to their highly dynamic nature within the parasite cytosol (*Videos 5–7*), a relatively stable F-actin network was clearly visible that linked individual parasites within the vacuole. Chromobodies have been used successfully in diverse eukaryotes to analyse actin dynamics (*Panza et al., 2015*; *Plessner et al., 2015*; *Rocchetti et al., 2014*) and appear to have less toxicity than other F-actin sensors. However, we were concerned that Cb expression in *T. gondii* would perturb actin polymerisation, especially given the proposed dynamic and unstable nature of *T. gondii* actin. Indeed, high levels of transiently expressed Cb inhibited dense granule motility, likely by stabilizing F-actin (*Figure 3C*). However, upon stable expression of Cb-Halo or Cb-Emerald, no effect on the actin-dependent processes of invasion, replication and egress were observed, while only small alterations in dense granule movement and parasite motility were detectable (*Figure 4*). The reason for these discrepancies is currently unknown, but may be related to differing sensitivities of actin dynamics for these processes. Additionally, we demonstrated that the total amount of F-actin in the parasite does not change upon expression of Cb (*Figure 5*), indicating that while actin dynamics may be modulated, the overall polymerization state of actin does not change. We show that parasite actin remains susceptible to depolymerising and stabilizing agents Cyt-D and Jas. Moreover, the localization of Cb to filamentous structures in the residual body and inter-parasite network critically depends on the presence of actin (*Figure 6*). These data confirm that genetically encoded Cb-Halo and Cb-Emerald does not significantly alter actin dynamics within the parasite and so is an appropriate and robust tool for examining F-actin in this system.

The presence of actin in the RB and inter-parasite tubules has not previously been observed or predicted, and so we sought to independently verify that this localization was not an artefact of Cb expression. Labelling parasites with α-actin that preferentially recognises F-actin (*Angrisano et al., 2012b*) allowed the detection of actin positive structures connecting individual parasites (*Figure 6*), though the stain appeared more diffuse. When actin filaments were stabilised by the presence of Jas (*Figure 6*) this stain becomes more prominent and this was not dependent on the expression of Cb. This suggests that F-actin is less accessible to exogenously added antibodies after fixation, perhaps due to epitope alterations during fixation or the association of actin binding proteins to filaments. Interestingly, while actin has never been previously associated with this membranous network in *T. gondii*, it appears very similar to structures recently described for the related parasite *Theileria annulata*, which was shown to contain F-actin in a similar configuration (*Kühni-Boghenbor et al., 2012*).

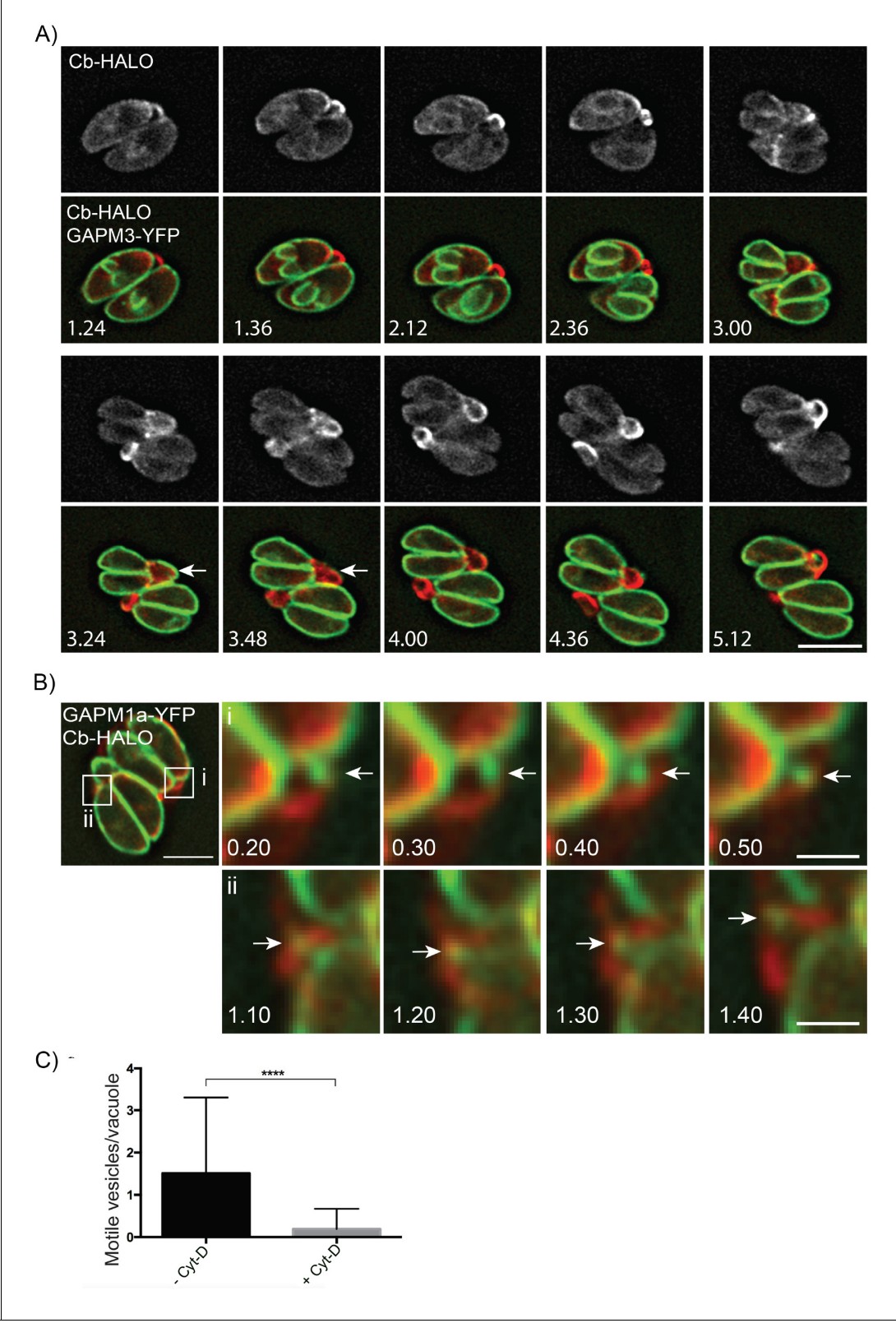

**Figure 8.** Parasite-derived extracellular vesicles are transported in an F-actin dependent manner. (**A**) Parasites co-expressing GAPM3-YFP and Cb-Halo were imaged every 6 min for 5 hr, F-actin can be seen initially connecting the basal end of the parasites before accumulating beneath the forming daughter cells where it appears to concentrate towards the rear of the new daughters during emergence and recycling of the maternal IMC. Note the sudden collapse of the mother IMC into vesicles that appear to move towards the IMC of the nascent daughter cells (arrow). Scale bar 10 μm. (**B**) In

*Figure 8 continued on next page*

*Figure 8 continued*

parasites endogenously expressing GAPM1a-YFP, extracellular vesicles could be observed in close apposition to Cb-Halo labelled filaments (arrow). Parasites expressing Cb-Halo were imaged every second for up to 5 min. Extracellular vesicles positive for GAPM1a-YFP were observed to move along F-actin filaments (arrows). Scale bar, 5 μm. (C) The number of vesicles per vacuole that moved within 5 min of imaging were quantified in the presence and absence of 500 nM Cyt-D. At least 60 vacuoles were counted over three independent experiments. ****p<0.0001.

Using a combination of imaging and reverse genetics, we demonstrate that F-actin is required for the formation and/or maintenance of the residual body. Parasites lacking actin do not contain a residual body and are disorganized within the PV. In addition, we show that these extracellular actin-containing structures are required for the transport of material between parasites within a vacuole. The residual body and inter-parasite network allow both the free diffusion of cytoplasmic material between parasites, and transport of membrane bound vesicles between parasites. Further work will be required to determine if these membrane bound vesicles are then taken up by other parasites or are simply transported to the RB to remove them and if a myosin motor powers this for its transportation, as is the case of dense granule transport within the parasite cytosol (*Heaslip et al., 2016*).

The organization of F-actin within the parasites and vacuole is highly dynamic throughout the cell cycle (*Figures 8* and *9*). In interphase parasites, long inter-parasite contentions extend throughout the PV. At the beginning of the replication cycle this network collapses and is found concentrated in the residual body (or bodies). Within the parasites, F-actin is found concentrated at the IMC of growing daughter cells during elongation (*Figures 8* and *9*). As the daughters bud from the mother cell, the inter-parasite connections again extend throughout the PV. As would be expected, the network also collapses in response to calcium ionophore, freeing the parasites and allowing egress. As motile parasites leave the PV, actin appears concentrated at the basal end of parasites (*Figures 8* and *9*).

While Cb expression allowed us to demonstrate F-actin-containing membranous tubules for the first time, we were unable to assess the length of the individual F-actin filaments using these techniques. We speculate that this network consists of either short F-actin bundles that are cross-linked via unknown actin-binding proteins (based on the formation of short actin filaments in vitro) (*Skillman et al., 2011*) or that the presence of actin binding proteins such as formin, profilin, and coronin may coordinate their activities in vivo to produce longer actin filaments than those formed in vitro (*Skillman et al., 2011*; *Olshina et al., 2016*;*Salamun et al., 2014*). While this study focuses on

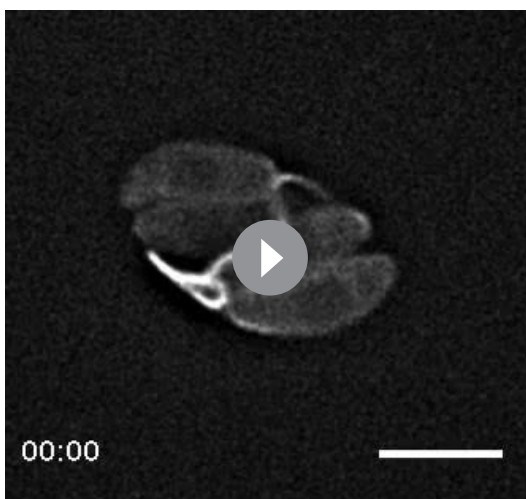

**Video 8.** Analysis of Cb-Halo during two rounds of replication. Images were taken every 30 min for 20 hr. The network appears dynamic across the intracellular lifecycle, collapsing into rings during daughter cell emergence before reforming. Scale bar 5 μm.

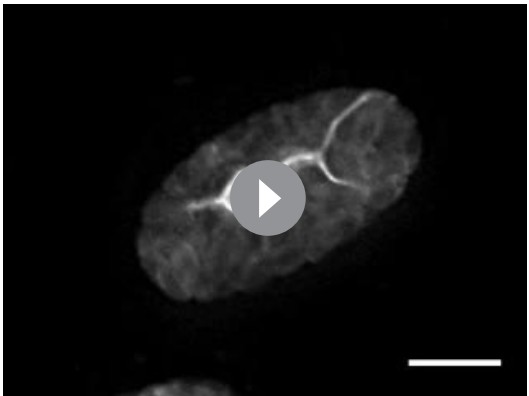

**Video 9.** Imaging of F-actin dynamics after addition of Ca2+-Ionophore, images taken every second. Filaments break up in a calcium dependent manner before parasites start to egress.

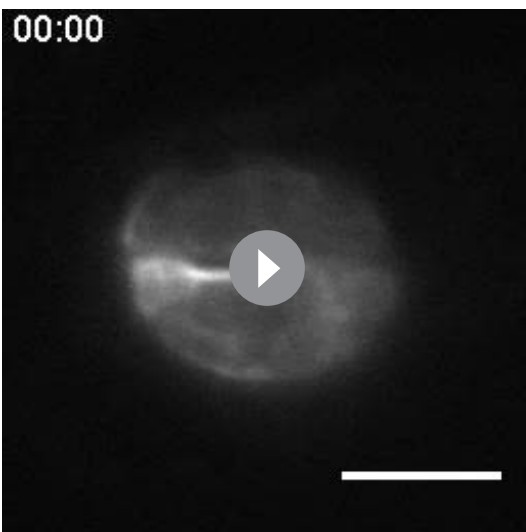

**Video 10.** FRAP of cytosolic filaments. FRAP treatment in cells stably expressing Cb-Emerald. After FRAP treatment, the F-actin inside the cell shows a fast fluorescence intensity recovery time of 20 s. Scale bar 5 µm. Imaging speed 5 fps.

**Video 11.** FRAP of filamentous structure. FRAP treatment in cells stably expressing Cb-Emerald. After FRAP treatment in the nanotubular network, F-actin shows a fluorescence intensity recovery time of 60 s. Scale bar 5 µm. Imaging speed 5 fps.

the characterization of the membranous network within the PV, it is worth nothing that highly dynamic actin filaments have been also detected within the cytosol of the parasite, and we show that these dynamics are almost completely abolished upon depletion of ADF in good agreement with previous findings (*Haase et al., 2015*; *Mehta and Sibley,*

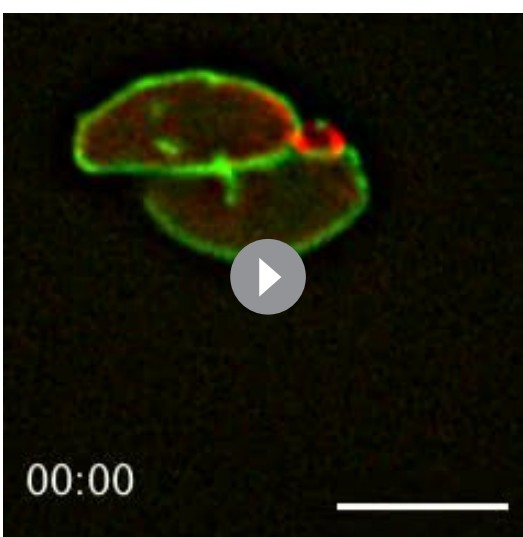

**Video 12.** GAPM3-YFP expressing parasites expressing Cb-Halo were imaged every 6 min for 5 hr, F-actin can be seen initially connecting the basal end of the parasites before accumulating beneath the forming daughter cells where it appears to concentrate towards the rear of the new daughters during emergence and recycling of the maternal IMC. Scale bar 5 µm.

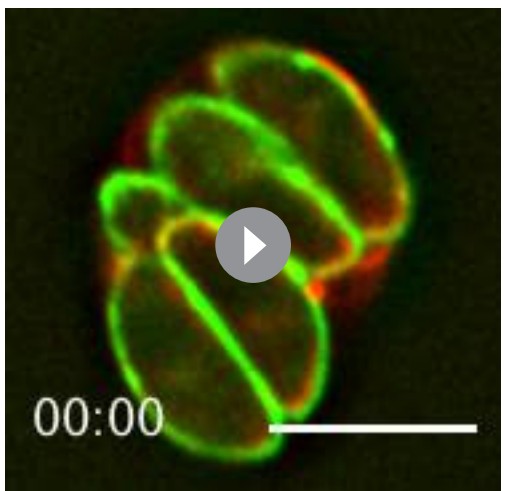

**Video 13.** Vesicle tracking on F-actin tubules. Parasites stably expressing GAPM1a-YFP and transiently expressing Cb-Halo were imaged every second and vesicles containing GAPM1a-YFP could be observed to move along Cb-Halo filaments. Time indicated in minutes, scale bar 5 µm.

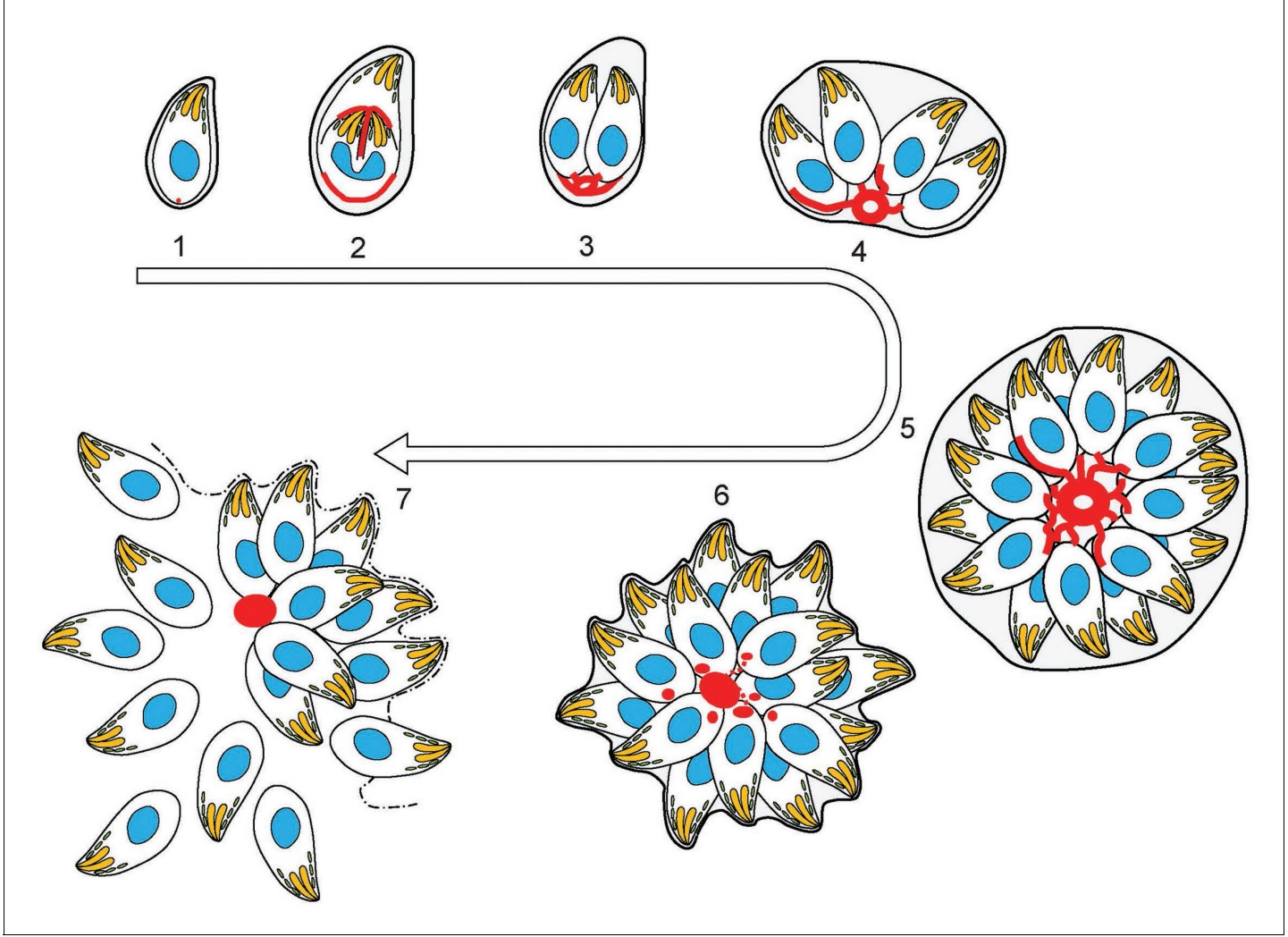

**Figure 9.** (1) After successful invasion, tachyzoites establish a parasitophorous vacuole and initiate replication. (2) During daughter cell formation, actin labelling is observed at the IMC of the daughter cells and at the posterior pole of the mother. (3) Once the daughter cells are fully formed, the actin signal strongly localises at the posterior end of the parasites and with the remains of the mothers IMC, as it is recycled. The first actin filamentous network and ring-like structures are visualized. (4–5) Replication continues and the filamentous network is established between the tachyzoites. The actin ring continues to localize at the residual body. (6) The filaments between the parasites and the ring break in a calcium dependent manner prior to egress. The network collapses and dots of actin are detected at the posterior end of tachyzoites. (7) Tachyzoites egressing from the vacuole leave behind an accumulation of actin in the residual body.

*2011*) (*Figure 6*; *Videos 5–7*). Defining the organization of cytosolic filaments will be critical to further elucidating the role of F-actin both during intracellular processes such as vesicle transport and extracellular processes such as motility and invasion, where the role of the parasites acto-myosin system is currently readdressed, since parasites devoid of detectable actin remain invasive (*Whitelaw et al., 2017*). Interestingly, it appears that host cell membrane dynamics, driven by host cell actin, appears to be modulated by the parasite, enabling invasion in the absence of a functional parasite acto-myosin system (*Bichet et al., 2016*).

It is possible that F-actin is organized into distinct, higher order structures, perhaps forming large stable bundles in the tubules while existing as single filaments or small bundles in the cytosol. Future work will be needed to further define the organization of the actin cytoskeleton and to identify the actin binding proteins which contribute to the formation of these structures.

## Materials and methods

### Plasmid construction

The Cb-Halo plasmid consists of a sequence encoding actin chromobody (Cb) from Chromotek followed downstream by an in frame sequence encoding Halo (Promega). The vector backbone contains a *Toxoplasma* tubulin promoter for protein expression and *hxgprt* resistance cassette. Actin-Cb was amplified with primers FW pG1Cb atta<u>GAATTC</u>CCTTTTTCGACAAA<u>ATG</u>GCTCAGGTGCAGCTGGT and Rv pG1Halo TATGTTAATTAATTAACCGGAAATCTCCAGAGTAG using as a template pHTC Halo Tag (Promega) containing in frame actin Cb. Actin chromobody (actin-Cb) was cloned in frame into pHTC Halo Tag (Promega) using a PCR product generated with primers FWvhH <u>GAATTC</u>ATGGCTCAGGTGCAGCTGGTGGA, RVvhH <u>CTCGAG</u>GCTTCTTGAGGAGACGGTGACCT using a pAC-TagRFP (Chromotek) as a template. To endogenously tag *gapm1a*, the 3'flank of the gene was amplified using 5' TACTTCCAATCCAATTTAATgccgccctgttcgtgtagtttatctg 3' and 5' TCCTCCACTTCCAATTTTAGCGGATCTGCAGGACAGGCAAGCC 3' and inserted into LIC-YFP by ligation independent cloning (*Huynh and Carruthers, 2009*). To create Cb-EmFP, the emeraldFP coding sequence was amplified using primers (emeraldFP-F: atgcaccggtatgggactcgtgagcaaggg and EmeraldFP-R: atgccttaagttacttgtacagctcgtcca). The emeraldFP PCR product and Cb-Halo plasmid were subcloned using traditional restriction digestion/ligation protocols.

### Culturing of parasites and host cells

Human foreskin fibroblasts (HFFs) (RRID: CVCL_3285, ATCC) were grown on tissue culture-treated plastics and maintained in Dulbecco's modified Eagle's medium (DMEM) supplemented with 10% foetal bovine serum, 2 mM Lglutamine and 25 mg/mL gentamycin. Parasites were cultured on HFFs and maintained at 37°C and 5% CO2. Cultured cells and parasites were regularly screened against mycoplasma contamination using the LookOut Mycoplasma detection kit (Sigma) and cured with Mycoplasma Removal Agent (Bio-Rad) if necessary.

### *T. gondii* transfection and selection

To generate stable Cb-Halo expressing parasites, $1 \times 10^7$ of freshly released RH $\Delta hxgprt$ parasites were transfected with 20 µg DNA by AMAXA electroporation. Selection was based on mycophenolic acid and xanthine (*Donald et al., 1996*). Gapm1a-YFP parasites were transfected as above and were selected using pyrimethamine. In order to express *adf* cKO stably expressing Cb-Emerald, parasites were transfected with Cb-Emerald and selection was performed using flow cytometry with a S3 Cell Sorter (Bio-Rad, Hercules, CA, USA).

### Inducing the conditional *act1* cKO

The inducible *act1* cKO was obtained by the addition of 50 nM rapamycin to the parental LoxPAct1 strain for 4 hr at 37°C, 5% $CO_2$ and cultured as described in *Egarter et al. (2014)*.

### Cb-Halo expression in the *act1* cKO

Cb-Halo plasmid was transiently transfected into LoxPAct1. Transfected parasites were induced with 50 nM rapamycin for 4 hr (*Andenmatten et al., 2013*), washed and plated on HFFs. Parasites were fixed at the desired time and stained with Halo-TMR (1:10,000) for 15 min.

### Western blot

Extracellular parasites were pelleted and then resuspended in RIPA buffer (50 mM Tris–HCl pH 8; 150 mM NaCl; 1% Triton X–100; 0.5% sodium deoxycholate; 0.1% SDS; 1 mM EDTA), incubation for 5 min on ice was used to lyse the cells. Afterwards, samples were centrifuged for 60 min at 14,000 rpm at 4°C and laemmli buffer was added to the supernatant. $5 \times 10^6$ parasites were loaded onto an SDS acrylamide gel. Western blotting was performed as described previously (*Herm-Götz et al., 2007*) using IRDye680RD or IRDye800RD (Li-Cor) secondary antibodies.

### Co-immunoprecipitation

Extracellular Wt and Cb-Halo parasites were harvested, filtered and washed before being resuspended in actin stabilization lysis buffer (60 mM PIPES, 25 mM HEPES, 10 mM EDTA, 2 mM MgCl₂,

125 mM KCl completed with Pierce Protease inhibitor mini tablets, EDTA Free (Thermo Scientific) and Triton X-100 0.2%). Lysates were incubated on ice for 1 hr, then incubated with equilibrated Magne HaloTag Beads (Promega) for 2 hr at 4°C. Beads were washed 5 times with 1 ml of buffer and elution was made using the TEV protease (Promega) as instructed in the protocol. Western blot analysis was performed as above.

## F-Actin stabilization experiment

Freshly egressed parasites were harvested and resuspended in buffer A (60 mM PIPES, 25 mM HEPES pH7.5, 10 mM EDTA, 2 mM MgCl2, 125 mM KCl) containing 1 µM Jasplakinolide or DMSO. Parasites were incubated for 1 hr at 37°C in a water bath. After centrifugation, parasite pellets were resuspended in buffer B (buffer A complemented, 10% glycerol and 1% Triton X-100). The suspensions were left on ice for 1 hr and centrifuged at 13,000 rpm for 30 min at 4°C. Pellets were washed once with buffer B, resuspended in SDS protein loading buffer, and boiled. Western blot analysis and semi quantification was performed as described above using Li-Cor Odysseys Clx with antibodies against ACT1 (*Angrisano et al., 2012b*) and GRA7 as a loading control.

## Purification of rCb from bacteria

The Cb coding sequence with a C-terminal 6-His tag was cloned into a pET22b bacterial expression vector and transfected into chemically competent Rosetta (DE3) bacteria (EMD Millipore). A 100 ml culture of LB-ampicillin was grown for 24 hr from a single bacterial colony. 25 mls of bacteria was used to inoculate 500 mls of LB-ampicillin and grown at 37°C until OD between 0.6 and 0.8. Expression was induced with 0.5 µM IPTG at 37°C for 4 hr. Bacterial pellets were frozen overnight at −80°C. Bacteria pellets were resuspended in 40 mls of xTractor buffer (Clontech) and then sonicated for 4 min on ice. Extracts were clarified at 9500x g for 20 min at 4°C. Supernatant were added to 1 ml of equilibrated Talon resin and agitated at 4°C for 60 min. Supernatant/resin mix was added to affinity column (Biorad) and supernatant allowed to flow through by gravity. Resin was washed with 20mls of Talon equilibration buffer (Clontech) followed 10 mls wash buffer (Equilibration buffer with 1/10th volume of elution buffer). Proteins were eluted with 10mls of elution buffer in 1 ml aliquots. Elutions containing rCb were pooled and dialyzed overnight in 125KMEI buffer (125 mM KCl, 1 mM MgCl$_2$, 1 mM EGTA, 10 mM Imidazole pH 7.0, 10 mM DTT). All steps were performed at 4°C and proteins were stored at 4°C. Protein concentration was determined using Bradford Assay.

## rCb affinity assays

rCb was diluted to 20 µM in 125KMEI and clarified at 100,000xg for 20 min at 4°C. Protein concentration in supernatant was determined using Bradford Assay and diluted to 8 µM in 125KMEI. Actin was diluted to various concentrations in Actin Buffer (25 mM KCL, 1 mM EGTA, 25 mM imidazole pH 7.4, 4 mM MgCl$_2$, 10 mM DTT). Actin and rCb were added together in equal volumes and incubated at room temperature for 30 min before centrifugation at 100,000xg for 20 min at 4°C. Equal volumes of supernatant and pellet were run on NuPAGE Bis-Tris 4–12% Gradient gels (ThermoScientific) with 1XMES buffer. Gels were stained using Simply Blue Coomassie Stain as per manufacturer's instructions. The ratio of rCb in supernatants and pellets was determined by densitometry using ImageJ (RRID: SCR_003070). The data from two independent experiments were used to determine the apparent Kd of rCb for F-actin.

## Treatment with actin remodelling compounds

Parasites were incubated with either 100 nM jasplakinolide or 2 µM cytochalasin D for 1 hr at 37°C. Parasites were fixed with 4% PFA and counterstained with the respective antibodies.

## Light microscopy

Widefield images were acquired in z-stacks of 2 µm increments and were collected using an Olympus UPLSAPO 100× oil (1.40NA) objective on a Deltavision Core microscope (Applied Precision, GE) attached to a CoolSNAP HQ2 CCD camera. Deconvolution was performed using SoftWoRx Suite 2.0 (Applied Precision, GE). Video microscopy was conducted with the DeltaVision Core microscope as above. Normal growth conditions were maintained throughout the experiment (37°C; 5% CO$_2$). Further image processing was performed using ImageJ64 software. FRAP data were recorded using the

same microscope as above. The region of interest was photobleached with a 405 laser for an optimised number of events for the cell strain and area investigated. Three pre-bleach and a number of post-bleach images ranging between 20 and 180 s (one image per second) were recorded with Exc and Em filter for FITC with an exposure time of 100–200 ms, ND filter 32%. Data were displayed and analysed using ImageJ software (*Sultana et al., 2007*). Fluorescence intensity was expressed as intensity percentage of the same unbleached area (filament or nanotubular network) to account for photobleaching and defocussing in the sample. Super-resolution microscopy (SR-SIM) was carried out using an ELYRA PS.1 microscope (Zeiss) as described in *Harding et al. (2016)*. For filament size analysis default settings of the Ridge Detection Plug-In (*Steger, 1998*) in ImageJ was used.

## Phenotypic characterisation of Cb-Halo parasites

### Plaque assay

Conducted as described in *Meissner et al. (2002)*. $1 \times 10^3$ parasites were inoculated on a confluent layer of HFF and incubated for 5 days, after which the HFF were washed once with PBS and fixed with ice cold MeOH for 20 min. HFFs were stained with Giemsa with the plaque area measured using ImageJ. Mean values of three independent experiments ± SEM were determined.

### Dense granule motility

Cb-Halo was transfected into SAG1ΔGPI-GFP expressing parasites, allowed to invade a confluent HFF monolayer on grown on MatTek (MatTek Corporation, Ashland, MA) dishes overnight, labelled with Halo-TMR was described above and dense granule motion was imaged as previously described (*Heaslip et al., 2016*). Directed dense granule motions were tracked using the ImageJ Plug-in MTrackJ and the number of directed runs/parasite/minutes and directed run-lengths was quantified. The TMR fluorescence was measured using ImageJ and correlated to the dense granule directed motions. Total Number of directed runs in control, low expression and high expression samples were 183,150 and 1 respectively. Total number of vacuoles analysed from control, low expression and high expression were 19,17 and 14 respectively from two independent transfections. Statistical method used was student's t-test.

### Trail deposition assay

Gliding assays were performed as described before (*Håkansson et al., 1999*). Briefly, freshly released parasites were allowed to glide on FBS-coated glass slides for 30 min before they were fixed with 4% PFA and stained with α-SAG1 under non-permeablising conditions. Mean values of three independent experiments ± SEM were determined.

### Invasion/replication assay

For the assay $5 \times 10^4$ freshly released parasites were allowed to invade a confluent layer of HFFs for 1 hr. Subsequently, five washing steps were performed for removal of extracellular parasites. Cells were then incubated for a further 24 hr before fixation with 4% PFA. Afterwards parasites were permeabilised and stained with α-IMC1 antibody (*Egarter et al., 2014*). For invasion the number of vacuoles in 15 fields of view were counted. For replication, 200 vacuoles were counted for the number of parasites per vacuole. Mean values of three independent experiments ± SEM were determined.

### Quantification of replication of *act1* cKO

LoxP*act1* parasites were induced with 50 nM rapamacyn for four hours. *LoxPact1 were used* as control. Parasites were allowed to invade in HFF cells grown on glass coverslips 24, 48, 72 and 96 hr after induction and replication was analysed 24 hr later. Coverslips were fixed using 4% paraformaldehyde for 20 min at room temperature and mounted using DAPI Fluoromount-G (SouthernBiotech). Nuclei staining was used to determine the number of parasites per vacuole.

### Quantification of actin filaments in *act1* cKO

LoxP*act1* parasites expressing Cb-Halo were induced with 50 nM rapamycin for four hours. Cb-halo expressing RH Δ*hxpgrt* parasites were used as control. Parasites were allowed to invade HFF cells grown on coverslips and replicate for 24, 48, 72 and 96 hr. Coverslips were exposed to Halo-TMR

for 15 min prior fixation. Then fixed using 4% paraformaldehyde for 20 min at room temperature. Only vacuoles with more than eight parasites (when filaments are clearly visible) were considered for quantification.

## Egress assay

Egress assays were performed as described in *Black et al. (2000)*. Briefly, $5 \times 10^4$ parasites were grown on HFF monolayers for 36 hr. Media was exchanged for pre–warmed, serum–free DMEM supplemented with 2 µM A23187 (in DMSO) to artificially induce egress. After 5 min the cells were fixed with 4% PFA and stained with a-SAG1 antibody under non-permeabilising conditions. 200 vacuoles were counted for their ability to egress out of the host cells. Mean values of three independent assays ± SEM were determined.

## Live egress

Cb-Halo parasites were prepared akin to the egress assay. Around $1 \times 10^5$ parasites were incubated in an Ibidi µ-Dish$^{35\ mm}$, high and left to replicate for 36 hr. Halo-TMR ligand (1:5000) was added to the dish, washed out after 15 min. The dish was then transferred to the DeltaVision Core microscope (Applied Precision, GE) with standard growth conditions. 20 µM $Ca^{2+}$ ionophore was added to the media after imaging had commenced. Images were captured at 1 frame per second using the Soft-WoRx software. Further image processing was performed using Fiji software.

## Live cell invasion

Parasites were artificially released using 23 G needle and filtered prior to inoculation on a confluent layer of HFFs, grown on glass bottom dishes. The dish was then transferred to the DV Core microscope (Applied Precision, GE) and maintained under standard culturing conditions. Images were captured at 1 frame per second in DIC using a 40x objective lens. Images were analysed using the Fiji software for point of entry to closure. Penetration speeds were obtained for 22 independent invasion events for both RH and Cb-Halo parasites.

## Live cell 2D motility

Both RH and Cb-Halo parasites were artificially released using a 23 G needle and filtered, spun down and resuspended in pre-warmed gliding buffer (1 mM EDTA, 10 mM HEPES in HBSS). These were then added onto FBS-coated glass bottom dishes and transferred to the DV core microscope (Applied Precision, GE). The cells were maintained under standard culturing conditions and imaged at one image per second using a 20x objective lens. Image sequences were analysed using the Fiji software, plugin wrMTrck. Average distance and speed were calculated for 20 parasites exhibiting helical or circular motions. Statistics were analysed using GraphPad Prism7.0.

## Scanning electron microscopy

Infected cells were fixed in 2.5% glutaraldehyde and 4% paraformaldehyde in 0.1 M phosphate buffer. Following several washes with 0.1 M phosphate buffer, the cells were dehydrated in ascending ethanol series and critical point dried (Tousimis, USA). Before metal sputtering, the cell monolayer was scraped with Scotch tape, exposing the cytoplasm of the cells, as well as the parasitophorous vacuoles. These exposed cells were metal coated with gold/palladium and observed in a Jeol 6400 scanning electron microscope (Jeol, Japan).

## Correlative light-electron microscopy (CLEM)

Cells were grown in gridded glass bottom petri dishes (MatTek) and infected with Cb-Halo parasites. Vacuoles presenting an extensive intravacuolar network were imaged with SR-SIM in an ELYRA PS.1 microscope (Carl Zeiss, Germany), and the material was fixed in 2.5% glutaraldehyde and 4% paraformaldehyde in 0.1 M phosphate buffer; and processed for transmission electron microscopy as described previously (*Loussert et al., 2012*). Thin sections of the same areas imaged in 3D-SIM were imaged in a Tecnai T20 transmission electron microscope (FEI, Netherlands). For correlative light/cryo-electron microscopy, cells infected with Cb-Halo parasites were fixed in 4% paraformaldehyde and 0.2% glutaraldehyde in 0.1 M phosphate buffer, pH 7.2, infiltrated in 2.1 M sucrose overnight and rapidly frozen by immersion in liquid nitrogen. Cryo-sections were obtained at −100°C using an

Ultracut cryo-ultramicrotome (Leica, Austria). Cryo-sections were blocked in 3% bovine serum albumin in phosphate buffer and incubated in the presence of anti-chromobody. After several washes in blocking buffer, the cryo-sections were imaged in an Elyra super-resolution microscope (Carl Zeiss, Germany) and then incubated with 10 nm, gold-labelled anti-protein A (Aurion, Netherlands). The same areas observed on the light microscope were imaged in a Tecnai T20 transmission electron microscope (FEI, the Netherlands).

## Acknowledgements

We would like to thank Gary Ward (University of Vermont) for the IMC1 antibody, Jake Baum (Imperial College, London) and Dominique Soldati (University of Geneva) for ACT1, GAP40 antibodies. Freddy Frischknecht and Isabelle Tardieux for critically reading the manuscript. Special thanks to Dr. Gurman Pall and other lab members for stimulating discussions. CRH is supported through a Sir Henry Wellcome Fellowship (WT103972AIA). This work was supported by an ERC-Starting grant (ERC-2012-StG 309255-EndoTox), the Wellcome Trust 087582/Z/08/Z Senior Fellowship for MM and National Institutes of Health R21 grant awarded to AH (AI121885). The Wellcome Trust Centre for Molecular Parasitology is supported by core funding from the Wellcome Trust (085349). The funders had no role in study design, data collection and analysis, decision to publish, or preparation of the manuscript. The authors have declared that no competing interests exist.

## Additional information

### Funding

| Funder | Grant reference number | Author |
| --- | --- | --- |
| Wellcome | 087582/Z/08/Z | Markus Meissner |
| H2020 European Research Council | ERC-2012-StG 309255-EndoTox | Markus Meissner |
| Wellcome | WT103972AIA | Clare Harding |
| National Institute for Health Research | AI121885 | Aoife Heaslip |

The funders had no role in study design, data collection and interpretation, or the decision to submit the work for publication.

### Author contributions

JP, AH, Conceptualization, Data curation, Formal analysis, Investigation, Visualization, Methodology, Writing—original draft, Writing—review and editing; JW, Data curation, Formal analysis, Investigation, Visualization, Methodology, Writing—original draft; CH, Conceptualization, Data curation, Formal analysis, Supervision, Funding acquisition, Validation, Investigation, Visualization, Methodology, Writing—original draft, Writing—review and editing; SG, MAR, Investigation, Methodology; MIDRM, Data curation, Formal analysis, Investigation, Methodology; FL-B, Formal analysis, Investigation, Methodology; LL, Investigation, Visualization, Methodology, Writing—original draft; RI, Conceptualization, Methodology, Writing—original draft, Writing—review and editing; MM, Conceptualization, Resources, Formal analysis, Supervision, Funding acquisition, Validation, Investigation, Visualization, Methodology, Writing—original draft, Project administration, Writing—review and editing

### Author ORCIDs

Markus Meissner, http://orcid.org/0000-0002-4816-5221

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
