## [Decision Letter]

Thank you for submitting your article "*Toxoplasma gondii* F-actin forms an extensive filamentous network required for material exchange and parasite maturation" for consideration by *eLife*. Your article has been reviewed by two peer reviewers, and the evaluation has been overseen by Anna Akhmanova as the Senior and Reviewing Editor. The following individuals involved in review of your submission have agreed to reveal their identity: Isabelle Tardieux (Reviewer #2).

The reviewers have discussed the reviews with one another and the Reviewing Editor has drafted this decision to help you prepare a revised submission.

Summary:

This manuscript addresses the organization of actin filaments in the protozoan *Toxoplasma gondii*. The authors use an actin probe based on camelid antibodies (chromobody, Cb) to trace endogenous actin during intracellular life cycle. This approach has already proved successful in monitoring actin dynamics in various systems ranging from mammalian cell lines to the whole zebrafish. This approach is highly relevant in the context of *Toxoplasma* and related parasites, since the understanding of actin dynamics (i.e. assembly/disassembly/2D and 3D organization) remains quite elusive in this organism and the processes of formation and turnover of polymerized actin pool are subject of controversy. Use of a chromobody helps to avoid the drawbacks of direct tagging or overexpression. This study clearly brings novelty to the field but also challenges the existing data in particular concerning the amount of F-actin, localization and network architecture in *Toxoplasma*. While the authors do not yet provide mechanistic explanations on how a particular F-actin bundle-like network that spreads in the vacuolar space, could nucleate, maintain the polarity of the progeny within the vacuole and possibly assist daughter cell individualisation, the first identification of this actin-based population provides a new angle for future dissection of the features of actin dynamics during *Toxoplasma* replication and possibly for uncovering additional F-actin sub-populations of distinct dynamics.

Essential revisions:

The strength of the tool/approach and the high informative value of some experiments (some live and static assays, especially the Figure 3, which is gorgeous) are somewhat weakened by inaccuracies in the text and also by limited sampling (quantitative data and statistics are not presented). To make this contribution stronger, we thus recommend to improve the clarity in the introductive part, to reorganize data presentation (the authors might consider showing the static study first, followed by the live study), to correct some editing errors throughout the text and to provide additional information as requested below. Please also check that for each figure presenting graphs, the definition of the error bars, the n numbers, the number of experiments and the statistical test used are included in the corresponding figure legends. Please also confirm that the statistical test used was appropriate (e.g., a t-test can be applied when the test statistic would follow a normal distribution).

1) The Abstract needs to be rewritten to help the reader on the rationale of the experimental approach (the actin chromobody approach is not mentioned) and the main results.

2) Abstract and Introduction:

*Toxoplasma* parasite needs to be introduced.

Egress needs to be introduced.

It is difficult to understand why the possible role(s) of actin described is/are unexpected? In addition, the basic reference to CC needs an explanation and for instance it should take into account the polar nature of the actin filament and therefore the intrinsic actin self-assembly and disassembly and the additional layer of extrinsic level of regulation by ABPs.

Introduction: Define briefly isodesmic actin model by comparison to conventional actin dynamics.

3) Results:

General remarks:

It is unclear whether Cb binds to F-actin, G-actin or both. It appears in some places to be argued one way and at times the other way. A normal anti-actin antibody staining would be expected to give background whereas Cb doesn't.

Furthermore, the authors may be correct that Cb may not adversely affect actin functions too much but they should be careful in interpreting this as evidence that it has no effect on actin polymerisation etc. in the cell. One might be affected without the other, and they have no direct measurement of actin in the cell.

Figure 1:

Please keep the same scale between WT and KO panels. It seems that the KO tachyzoites are longer despite they have a flat end. This is also visible in the Figure 2 (FRAP). If this is real, please comment this point.

Subsection “Depletion of actin results in the loss of the residual body”: If there is a loss of progeny synchronisation, it needs to be shown more convincingly, i.e. qualitatively and quantitatively (number tachyzoite/ vacuoles for n vacuoles and zoom staining). In Figure 1, the IMC labelling of the Act1cKO does not seem to support asynchrony but rather shows a problem of IMC biogenesis. In fact, the SEM (Figure 1) shows tachyzoites that have lost connection through the RB and are dispersed in the vacuolar matrix. The red arrows in SEM panels point to the end of division but it is not clear what the authors want to say. Do they mean the closure as individual cells (a step very poorly documented)?

Figure 2:

It should be explained why the bleached cell does not recover. It is proposed that GFP transport from surrounding cells is affected – but why is there no recovery from the newly synthesised GFP made in that cell?

FRAP assays: while only one pre-bleach panel is sufficient, the visualization for the FRAP effect should be improved. Delineating the FRAP area will also help. Why is there a range of exposure time (100-200 ms) for FRAP recording? Is it different from experiment to experiment? FRAP data should represent a certain number of experiments that should be mentioned. Could the authors clarify the following statement which as it is now, seems to lack real ground “Of note, digital tracking of vesicles suggested movement along a tubular or filamentous structure”.

Figure 3:

Panel B, on the right: the scale bar shows 10 micron unlike all the other figures.

Figure 5:

This figure is confusing as it does not bring strong support of actin in the RB with this imaging technique (it is even the opposite as the signal is everywhere despite the fact that the antibody is presented as kind of F-actin-specific). The chromobody characterized as a good marker of F-actin in the *Toxoplasma* vacuole (previous data) is much more convincing.

Figure 6:

This is a very important figure which would deserve to be better documented and clarified. Quantitative data on the network detection should be given since even in the jasplakinolide-treated cells, one vacuole containing 2 parasites seems not to display F-actin labelling (Figure 6)? Does that mean that the network is not always present at the same stage of parasite development in the vacuole? This is also puzzling for the loxActin (Figure 6) where few filaments (bundles as nicely seen later Figure 8) are observed among the tachyzoites in the vacuole. The description of the network that here looks as lying between tachyzoites within the vacuole is a little different from Figure 3, which shows a network in the connecting posterior structure throughout the replication cycle. This is to be partly answered with the Figure 8.

Figure 7 and the accompanying text:

Subsection “Cb specifically binds to parasite actin and does not alter the total amount of F-actin”. 2% of actin is in the F actin form: the authors might do a calculation (even a ballpark one) of the amount of actin they estimate as a percentage of the total protein, then the number of molecules and then the amount of F-actin that would be likely. Then look at images and see if reasonable. One might do it the other way as well. Does it really seem likely that the actin is as rare a protein as previous studies have suggested if 2% can be assembled into such extensive networks? It is hugely unlikely that MS proteomics detected *no* other proteins in the pull down. The statement “could not be identified" is ambiguous, please describe the results more clearly.

Figure 8:

What is the reason for qualifying the network as tubules (subsection “Inter-parasite actin tubules are dynamic during parasite replication and egress”)?

In the first paragraph of the aforementioned subsection. The filamentous actin is within tubules. If the authors are really sure that this is true – i.e. the correlative EM sections shown actually do fit in 3D with the LM images, this raises a really fundamental piece of biology that they must address in their discussion. How does actin (normally a cytoplasmic protein) get inside a membrane-lined tubule? We know a lot about how proteins cross membranes. However, here we have a claim that a protein that normally does not, does so in this system – and with all of its cohort proteins that facilitate filament formation and no doubt dynamics. This is not easy to imagine, so a discussion of how this might occur would be useful.

To improve the logic of presentation, the figure that shows the data related to the network dynamics during replication and the high-resolution description of the network by CLEM might be presented after Figure 3.

---

## [Author Response]

*Essential revisions:*

*The strength of the tool/approach and the high informative value of some experiments (some live and static assays, especially the Figure 3, which is gorgeous) are somewhat weakened by inaccuracies in the text and also by limited sampling (quantitative data and statistics are not presented). To make this contribution stronger, we thus recommend to improve the clarity in the introductive part, to reorganize data presentation (the authors might consider showing the static study first, followed by the live study), to correct some editing errors throughout the text and to provide additional information as requested below. Please also check that for each figure presenting graphs, the definition of the error bars, the n numbers, the number of experiments and the statistical test used are included in the corresponding figure legends. Please also confirm that the statistical test used was appropriate (e.g., a t-test can be applied when the test statistic would follow a normal distribution).*

We appreciate the suggestions, which allowed us to improve the clarity of the manuscript. In particular, we performed additional assays to quantify the amount of asynchronous replication of *act1cKO* parasites and added data for a conditional mutant for the actin depolymerisation factor (ADF) to further demonstrate formation of F-actin and F-actin dynamics in intracellular parasites. We considered to rearrange the data presentation, but felt that it would disrupt the “flow” of the study, since we organised data according to the characterisation of F-actin function (intracellular replication and formation of a F-actin network within the parasitophorous vacuole, transport of material in between individual parasites and egress).

The definition of error bars, n numbers, number of experiments and statistical tests are now included in the figure legends.

*1) The Abstract needs to be rewritten to help the reader on the rationale of the experimental approach (the actin chromobody approach is not mentioned) and the main results.*

We introduced the Chromobody in the Abstract.

*2) Abstract and Introduction:*

Toxoplasma parasite needs to be introduced.

*Egress needs to be introduced.*

We rewrote huge parts of the Introduction and elaborate on *Toxoplasma* and its asexual life cycle in more detail.

It is difficult to understand why the possible role(s) of actin described is/are unexpected? In addition, the basic reference to CC needs an explanation and for instance it should take into account the polar nature of the actin filament and therefore the intrinsic actin self-assembly and disassembly and the additional layer of extrinsic level of regulation by ABPs.

*Introduction: Define briefly isodesmic actin model by comparison to conventional actin dynamics.*

In our revision we tried to make the current discussion in the field clearer to the uninformed reader. However, it requires probably a full review to address all the implications and questions appropriately. We briefly introduced isodesmic vs. cooperative assembly mechanisms and the implication for actin binding proteins.

It now reads:

“According to the isodesmic polymerisation model, monomer addition is governed by a single equilibrium constant, meaning that no (unfavourable) activation step is required to initiate polymerisation. […] Furthermore, a recent study demonstrated that conditional depletion of actin in *T. gondii* results in complete abrogation of known actin functions, long before G-actin levels are fully depleted, suggesting that in vivothe formation of F-actin depends on a critical monomer concentration (Whitelaw, 2017 #1277).”

*3) Results:*

*General remarks:*

*It is unclear whether Cb binds to F-actin, G-actin or both. It appears in some places to be argued one way and at times the other way. A normal anti-actin antibody staining would be expected to give background whereas Cb doesn't.*

We made this clearer in the revision. The Cb itself is specific for F-actin, as shown in Figure 6, 7. The reason we obtain also a certain amount of cytosolic Cb-signal is that it is expressed within the parasite and so excess of CB will remain cytosolic. This can best be seen in Figure 6 for CytoD treatment or act1cKO parasites. Loss of F-actin filaments results in more cytosolic stain of parasites with CB. However, this is not due to G-actin binding, but due to cytosolic expression of CB.

We now mention “Filamentous structures were observed within the parasites. In addition, Cb was somewhat diffuse in the parasite cytosol, corresponding to unbound CB expressed in the cytosol of the parasite (Figure 3).”

And:

“By 48 hours, no F-actin structures could be observed and Cb-Halo was completely cytosolic, as expected due to cytosolic expression of this reagent (Figure 6).”

*Furthermore, the authors may be correct that Cb may not adversely affect actin functions too much but they should be careful in interpreting this as evidence that it has no effect on actin polymerisation etc. in the cell. One might be affected without the other, and they have no direct measurement of actin in the cell.*

We agree with the reviewers. Like all F-actin binding reagents, we would expect to influence actin dynamics. However, given that we can obtain parasites stable expressing CB and that the observed phenotypic consequences caused by this expression are relatively weak (though detectable) gives us confidence that our analysis of F-actin behaviour is correct.

We addressed this concern in the initial submission. To make it clearer to the reader we have now added:

“In summary expression of CB is well tolerated by the parasite indicating that it does not adversely affect actin functions in general. However, it cannot be ruled out that actin dynamics is locally affected within the cell due to expression of CB.”

*Figure 1:*

*Please keep the same scale between WT and KO panels. It seems that the KO tachyzoites are longer despite they have a flat end. This is also visible in the Figure 2 (FRAP). If this is real, please comment this point.*

We thank the reviewer for this comment. We corrected the size bars in Figure 1. In general we agree that there are morphological changes upon depletion of Act1 (as shown in Figure 1). However, the size of parasites is in general comparable.

*Subsection “Depletion of actin results in the loss of the residual body”: If there is a loss of progeny synchronisation, it needs to be shown more convincingly, i.e. qualitatively and quantitatively (number tachyzoite/ vacuoles for n vacuoles and zoom staining). In Figure 1, the IMC labelling of the Act1cKO does not seem to support asynchrony but rather shows a problem of IMC biogenesis. In fact, the SEM (Figure 1) shows tachyzoites that have lost connection through the RB and are dispersed in the vacuolar matrix. The red arrows in SEM panels point to the end of division but it is not clear what the authors want to say. Do they mean the closure as individual cells (a step very poorly documented)?*

We now quantified the loss of synchronisation and included it in Figure 1. We removed the Gra1 stain, since it is redundant to the Gra2 stain (a marker of the intravacuolar network). The IMC labelling in Figure 1 has been somewhat misinterpreted by the reviewer. We do not see any defects in early IMC formation during replication (see also Figure 9). However, the onset of replication within a PV is not synchronous. As an example, in Figure 1 a vacuole with 4 parasites is depicted, where each individual parasite is at a different stage during replication. The arrow depicts a parasite, where replication is almost complete, whereas other parasites are at an early stage of replication (or resting).

We also clarified the labelling (red arrow). We interpret the data as a defect in IMC maturation, leading to a collapsed posterior pole in good agreement with the live imaging analysis shown in Figure 9.

*Figure 2:*

*It should be explained why the bleached cell does not recover. It is proposed that GFP transport from surrounding cells is affected – but why is there no recovery from the newly synthesised GFP made in that cell?*

We would not expect a recovery of the signal due to GFP synthesis in the bleached cell, since we measure the recovery within 90 seconds. This is insufficient to detect novel protein synthesis.

*FRAP assays: while only one pre-bleach panel is sufficient, the visualization for the FRAP effect should be improved. Delineating the FRAP area will also help. Why is there a range of exposure time (100-200 ms) for FRAP recording? Is it different from experiment to experiment? FRAP data should represent a certain number of experiments that should be mentioned.*

We improved the depiction of the FRAP and chose a better movie for the control. The exposure time for both FRAP experiments was identical. In the initial submission we chose 2 different time scales, since recovery of wt parasites was very rapid (within 20 seconds), while act1cKO showed no recovery. We now chose the same time scale.

We also mention in the legend: “FRAP experiment is representative for several biological replicates (n>3)”.

*Could the authors clarify the following statement which as it is now, seems to lack real ground “Of note, digital tracking of vesicles suggested movement along a tubular or filamentous structure”.*

We are not sure what the reviewer refers to. The experiment we performed aimed to image transport of YFP-positive vesicles (Figure 2) in wt parasites (not expressing a CB). Tracking of these vesicles with tracking software suggested that they are transported along filaments. We included this experiment to demonstrate that this vesicle transport is seen irrespective of CB expression.

*Figure 3:*

*Panel B, on the right: the scale bar shows 10 micron unlike all the other figures.*

We updated this figure. We also included overview images of large PVs, where the intravacuolar network is identified (see comments below).

*Figure 5:*

*This figure is confusing as it does not bring strong support of actin in the RB with this imaging technique (it is even the opposite as the signal is everywhere despite the fact that the antibody is presented as kind of F-actin-specific). The chromobody characterized as a good marker of F-actin in the Toxoplasma vacuole (previous data) is much more convincing.*

We agree with the reviewers that this figure is somewhat distracting and we decided to remove it in the revision.

*Figure 6:*

*This is a very important figure which would deserve to be better documented and clarified. Quantitative data on the network detection should be given since even in the jasplakinolide-treated cells, one vacuole containing 2 parasites seems not to display F-actin labelling (Figure 6)? Does that mean that the network is not always present at the same stage of parasite development in the vacuole? This is also puzzling for the loxActin (Figure 6) where few filaments (bundles as nicely seen later Figure 8) are observed among the tachyzoites in the vacuole. The description of the network that here looks as lying between tachyzoites within the vacuole is a little different from Figure 3, which shows a network in the connecting posterior structure throughout the replication cycle. This is to be partly answered with the Figure 8.*

We thank the reviewer for these thoughtful comments. We now added additional data to demonstrate that:

The network that can be detected outside of the parasites (but still within membrane tubules) is indeed variable from vacuole to vacuole and depends on the size of the PV. It usually appears, once the PV contains at least 8 parasites. See also revised Figure 3. We added a new panel in Figure 3 to demonstrate this in more detail. The jasplakinolide treatment does result in actin polymerisation in all parasites, however the occurrence of the network is still variable, as seen in Figure 6. We now chose to show parasites that were transient transfected with Cb, since this allowed us to compare parasites expressing Cb and not expressing Cb in parallel. As can be seen in the larger panel, both parasites show identical staining with α-actin, irrespective of Cb expression. In case of *act1cKO* a network can never be observed irrespective of vacuole size, once actin is depleted (72-96hours post-induction). We now added quantifications for both assays.

We also decided to include our analysis of F-actin dynamics in a conditional knockdown for TgADF (Mehta et al., 2011) and demonstrate that F-actin dynamics is completely abrogated and F-actin accumulates at the RB. Importantly CB signal in the cytosol is almost completely lost, providing further evidence that CB is specific for F-actin.

*Figure 7 and the accompanying text:*

*Subsection “Cb specifically binds to parasite actin and does not alter the total amount of F-actin”. 2% of actin is in the F actin form: the authors might do a calculation (even a ballpark one) of the amount of actin they estimate as a percentage of the total protein, then the number of molecules and then the amount of F-actin that would be likely. Then look at images and see if reasonable. One might do it the other way as well. Does it really seem likely that the actin is as rare a protein as previous studies have suggested if 2% can be assembled into such extensive networks?*

We thank for this suggestion. We did indeed try to calculate and correlate amount of F-actin to signal. However, the complication here is that the coIPs can only be performed on extracellular parasites that do not form a network and where majority of actin appears non-polymerised (as published previously). This is in good agreement with our data, since in extracellular parasites CB-signal is diffuse and cytosolic, though filaments can be observed (see revised Figure 7 for an example of the signal in a freshly egressed, extracellular parasite). Unfortunately, co-IP on intracellular parasites is not feasible, since after preparation of the lysate the CB expressed within *Toxoplasma* is mixed with host cell lysate and hence we precipitate also huge amounts of host cell actin. Therefore, we cannot estimate at this point how much actin is polymerised during intracellular replication. The main reason, why we wanted to mention this result is to demonstrate that in extracellular parasites the expected amount of F-actin can be detected using CB, further demonstrating that expression of CB has minimal effects on F-actin dynamics.

*It is hugely unlikely that MS proteomics detected no other proteins in the pull down. The statement “could not be identified" is ambiguous, please describe the results more clearly.*

We agree that the initial description is somewhat misleading. We did indeed identify several proteins in this precipitation along with parasite actin. The important point however is that no host cell actin or any of the parasite actin-like and actin-related proteins was detected, demonstrating specificity for F-actin. We were contemplating to publish the whole list of identified proteins, which likely represent novel F-actin interacting proteins. However, at this point we would not like to share this set of data, since we are currently analysing them in detail.

*Figure 8:*

*What is the reason for qualifying the network as tubules (subsection “Inter-parasite actin tubules are dynamic during parasite replication and egress”)?*

*In the first paragraph of the aforementioned subsection. The filamentous actin is within tubules. If the authors are really sure that this is true – i.e. the correlative EM sections shown actually do fit in 3D with the LM images, this raises a really fundamental piece of biology that they must address in their discussion. How does actin (normally a cytoplasmic protein) get inside a membrane-lined tubule? We know a lot about how proteins cross membranes. However, here we have a claim that a protein that normally does not, does so in this system – and with all of its cohort proteins that facilitate filament formation and no doubt dynamics. This is not easy to imagine, so a discussion of how this might occur would be useful.*

We apologise for this apparent misunderstanding. The way we interpret the data is that the membranous tubules form a continuous system with the parasites/residual body, meaning that they have direct access to the parasites cytosol and therefore actin.

We now added in the Results section: “This situation appears very similar to the formation of tunnelling nanotubes, long filopodia like structures, which consists of thin F-actin-based membranous structures with a small diameter (20-500 nm) that facilitate long range communications between cells (Abounit, 2012 #127).”

*To improve the logic of presentation, the figure that shows the data related to the network dynamics during replication and the high-resolution description of the network by CLEM might be presented after Figure 3.*

We considered this suggestion. However, our logic is to first rule out artifacts (therefore we present first Figure 4 demonstrating minor influence of CB expression on the parasite, then specific binding of CB to F-actin (Figure 5, Figure 6), before we continue with the functional characterisation of the network. We are however happy to discuss other alternatives.